# A collaborative realist review of remote measurement technologies for depression in young people

Annabel E. L. Walsh ⑩ [1,2] ✉, Georgia Naughton ⑩ [3,7], Thomas Sharpe[3,7], Zuzanna Zajkowska[2], Mantas Malys[2], Alastair van Heerden ⑩ [4,5] & Valeria Mondelli ⑩ [2,6]

Digital mental health is becoming increasingly common. This includes use of smartphones and wearables to collect data in real time during day-to-day life (remote measurement technologies, RMT). Such data could capture changes relevant to depression for use in objective screening, symptom management and relapse prevention. This approach may be particularly accessible to young people of today as the smartphone generation. However, there is limited research on how such a complex intervention would work in the real world. We conducted a collaborative realist review of RMT for depression in young people. Here we describe how, why, for whom and in what contexts RMT appear to work or not work for depression in young people and make recommendations for future research and practice. Ethical, data protection and methodological issues need to be resolved and standardized; without this, RMT may be currently best used for self-monitoring and feedback to the healthcare professional where possible, to increase emotional self-awareness, enhance the therapeutic relationship and monitor the effectiveness of other interventions.

The use of digital technology is becoming increasingly common in mental health. Remote measurement technologies (RMT), such as smartphones and wearables, have been suggested as a solution to improve the monitoring, detection and treatment of depression in young people across the globe. However, the benefits, implementation and overall potential of RMT in the context of realist and ethical considerations remain unclear.

Depression is now a leading contributor to the global burden of disease with substantial impacts for the individual, society and economy (World Health Organization, WHO). It often begins in adolescence or young adulthood reflecting a developmentally sensitive period and has a chronic course with recurrent depressive episodes throughout one's lifetime[1,2]. Currently, monitoring and detection of depression relies on retrospective self-report influenced by both recall and mood-state associated biases[3]. Young people are particularly vulnerable to deterioration due to barriers in help-seeking, including delay in or lack of access to services, stigma and difficulty identifying or expressing concerns[4], often exacerbated by poor communication, disregard of agency[5], and mismanagement[6,7]. As such, many cases go undetected and untreated, contributing to a trajectory towards full-blown, chronic depression associated with poorer outcomes in adulthood[8]. Indeed, it has been estimated that only 28% of all individuals with

[1]The McPin Foundation, London, UK. [2]Department of Psychological Medicine, Institute of Psychiatry, Psychology & Neuroscience, King's College London, London, UK. [3]Young People's Advisory Group, The McPin Foundation, London, UK. [4]Centre for Community-based Research, Human and Social Capabilities Department, Human Sciences Research Council, Johannesburg, South Africa. [5]MRC/Wits Developmental Pathways for Health Research Unit, Department of Paediatrics, Faculty of Health Sciences, University of the Witwatersrand, Johannesburg, South Africa. [6]NIHR Maudsley Biomedical Research Centre, South London and Maudsley NHS Foundation Trust and King's College London, London, UK. [7]These authors contributed equally: Georgia Naughton, Thomas Sharpe. ✉e-mail: awalsh9890@gmail.com

depression receive treatment, which decreases to just 7–14% in low- and middle-income countries (LMICs)[9] where almost 90% of the global adolescent population reside[10]. For those who do receive treatment, clinical improvement is modest at best[11,12] with little understanding of the active ingredients of effective interventions and why they may work for some but not others[13–15]. There is a crucial need for effective early interventions targeting the adolescent period that can be personalized and are scalable across different contexts if we are to improve depression outcomes across the globe[16]. RMT may represent an increasingly ubiquitous and accessible resource for such an intervention.

RMT include any digital technology with the capability to collect data from an individual in real time during their day-to-day life through a remote interface, for example, smartphones, wearables and associated apps[17,18]. Data collection can be active via direct input by the individual (for example, ecological momentary assessment (EMA) and mood logs) and/or passive via embedded sensors or during interaction with the device (for example, heart rate, heart rate variation, skin conductance, actigraphy/accelerometery (method of measuring movement and sleep), GPS, ambient light, microphone, and paradata—screen status, app usage, and call/SMS logs)[17,18]. Extracted data features are then used to infer the mood, physiology, behaviour and environment of the individual (for example, stressful events and responses, rest/activity cycles, sleep, mobility, physical and social activity, and speech patterns), which may also be indicative of their mental state[17,18]. In this way, RMT could be used for real-time monitoring and detection of changes relevant to depression, contributing to more objective screening, improved symptom management and relapse prevention. Initial work is promising, with evidence that some RMT data features can distinguish those with depression from healthy controls and are associated with and predictive of standardized measures of depression symptom severity[19–21].

However, with pressure to find a scalable solution for the global mental health crisis and the movement further towards digital mental health, there is an ever-increasing danger that use of RMT may surge ahead of the evidence base[22]. Indeed, there is now an estimated 10,000 smartphone applications for mental health, but most are unregulated, have not been trialled for clinical effectiveness on depression outcomes, have low levels of engagement, and could even cause harm[23–26]. Depression itself, characterized by low motivation and energy, may make it more difficult to engage with RMT due to the requirement for daily interaction over long periods of time[27,28], and even less is known about the use of RMT for depression specifically in young people[29]. The benefits, successful implementation and overall potential of RMT are likely to depend on many different influencing factors. While narrative, systematic and scoping reviews have previously been conducted[30–34] and indicate the promise of RMT, they are limited in what literature can be included and are not well suited to developing in-depth understanding of these influencing factors and how RMT would work in the real world.

In this Article, as such, we conduct a realist review of RMT for depression in young people aged 14–24 years, involving iterative searches of a range of literature, collaboration with two young people co-researchers, and consultations with the McPin Young People's Advisory Group (YPAG). This approach will allow gradual refinement of an initial framework into an intervention theory describing the ways, for whom and in what contexts RMT appear to work or not work for depression in young people.

## Results

### Literature inclusion and characteristics

Of 6,118 records identified, 104 were included in the final evidence synthesis. Figure 1 is a flow diagram of search processes and record disposition, with Supplementary Data 1 providing details of all included records. Most investigated whether RMT data features could act as a proxy for depression symptom severity in young people (n = 36) and the acceptability and feasibility of doing so (n = 35). Data privacy ethics

(n = 6) or possible unintended outcomes (n = 3) of real-time monitoring were severely under investigated given the growth of the field. Eligible grey literature (n = 13) was limited mainly to clinical trial registrations, reflecting the surge in interest in the use of RMT for depression in young people but current lack of regulation and government policy. Only six records originated from LMICs (Nepal, Columbia, Bangladesh and Indonesia).

### What does and does not work?

**Type of RMT.** Out of the different types of RMT, smartphones and associated apps were the most prevalent in the literature (n = 52; internet-based platforms n = 23; wearables only n = 2; electronic diary n = 1; use in combination n = 22), as well as the most preferred by young people in the literature and during consultations, due to their ubiquity over other RMT[34,35]. However, the choice of RMT[36] and type of operating system used (that is, Android or iPhone)[37] were found to influence the specific data features identified, as well as overall accuracy, as a proxy for depression symptom severity, with wearables potentially being more accurate

**Passive versus active monitoring.** Regarding data collection, the continuous, unobtrusive nature of passive monitoring increased data quality and accuracy[36]; however, this still required the young person to remember to keep the device charged and on their person every day for extended periods, with no technological issues (for example, software crashes, global positioning system (GPS)/Wi-Fi/Bluetooth dropout, drain on battery and data allowances)[34,38]. As such, passive monitoring may be associated with overall lower adherence and greater decline in use over time compared with active monitoring[38].

Conversely, adherence to active monitoring required consideration of sampling frequency, but with conflicting evidence as to whether fewer or more EMA prompts was associated with greater adherence in young people with depression. While a meta-analysis found an association between more prompts per day and higher adherence in clinical samples[39], prompt fatigue at the daily level was common across studies, with more robust engagement obtained in studies employing weekly or biweekly EMA protocols[40,41]. It may be that the timing and content of prompts is more important than frequency, with the suggestion of smart prompts triggered when an individual is more likely to be receptive and able to respond[42], allowing convergence of prompt, motivation and ability as the three elements required for a behaviour to occur[43,44]. Provision of incentives, personalized feedback and gamification were also mentioned as ways to encourage continued engagement[45].

YPAG members expressed that both types of data collection should be used in combination for a holistic approach, with passive monitoring to increase accuracy, active monitoring to provide context, and interactions kept brief but relevant/personalized and motivational in nature, and of an appropriate frequency and timing to balance accuracy and intrusiveness. Young people in the literature and during consultations also expressed increasing concern regarding the privacy, confidentiality and ethical implications of the collection and sharing of digital data. Young people were much less likely to share digital data than psychosocial or biological data[46], with acceptability of certain digital data types particularly low, for example, audio, keystroke, social media, app usage and location data[46,47]. They stressed the importance of transparent, informed consent and the need for user control over what data are shared, how much, who with and when[48–50]. This would also allow the option to submit notes alongside for added context to avoid misinterpretation of the data or misrepresentation of the young person; however, one mentioned the risk of purposeful skewing of the data to reduce concern, for example, in co-morbid eating disorder.

**Accuracy as a proxy for depression.** Some RMT data features collected and assessed as potential proxies for depression symptom severity in young people included changes in sleep, mobility,

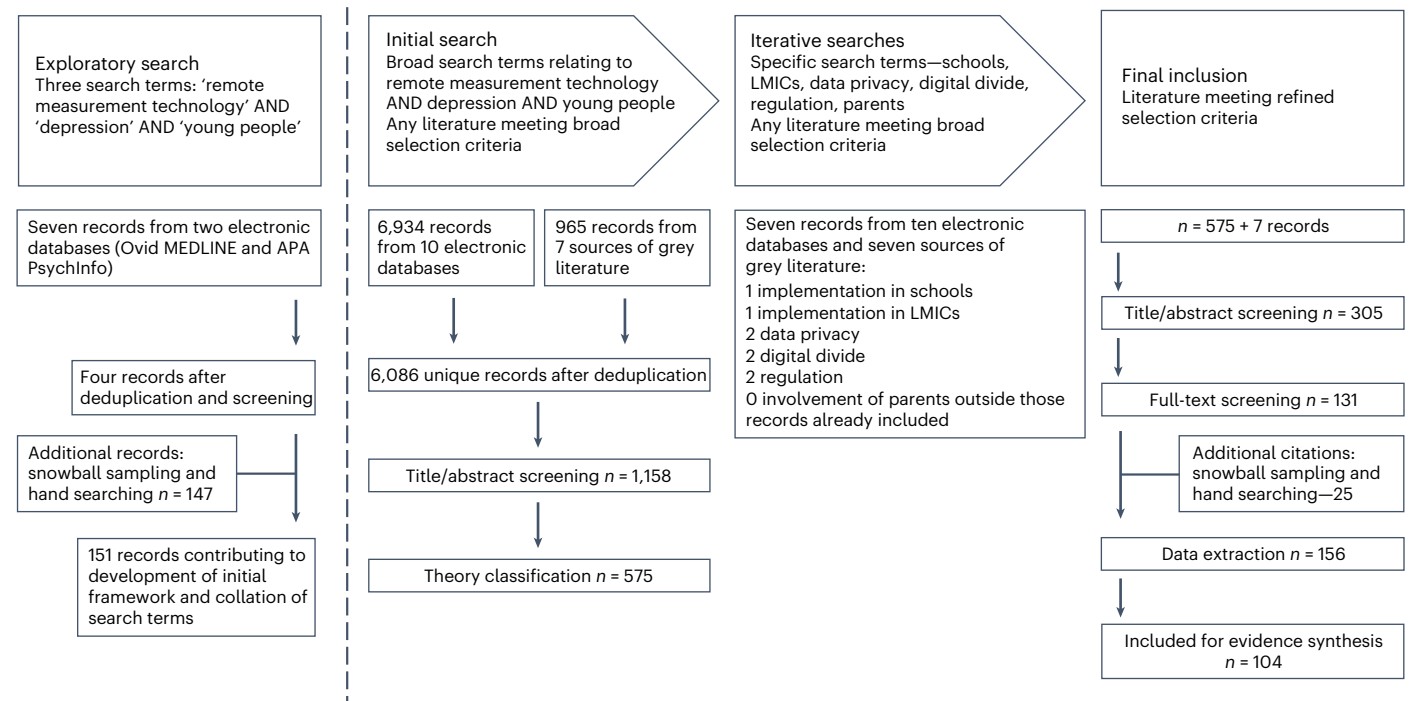

**Fig. 1 | Flow diagram of search processes and record disposition.** Shows the type of search and search terms used, and the number of records found and included from each search.

smartphone use, social communication and EMA responses. However, evidence was inconsistent as to whether these RMT data features could distinguish those with depression from healthy controls, or were associated with and predictive of standardized measures of depression symptom severity.

## Sleep

A case–control study found young people with depression had significantly longer sleep latency (time between going to bed and falling asleep), higher restlessness, shorter sleep duration and reduced sleep efficiency (ratio of total sleep time to time in bed) compared with healthy controls[51]. The longer the sleep latency, the greater the negative affect the following day[51]. Conversely, those recovered from depression had significantly longer sleep duration compared with healthy controls, specifically on work-free days[52]. This shows the importance of collecting contextual data, but also that sleep disruption may be a persistent factor in depression that would be sensible to monitor in relapse prevention. There was also conflicting evidence for the association between sleep duration and later depression symptom severity across longitudinal studies: one study found no significant correlation[53]; one found a negative correlation[54]; one found longer sleep duration at baseline but shorter sleep duration at follow-up associated with worse change in depression symptoms[55]; one found longer sleep duration to predict greater anhedonia the following day, but when averaged across 2 weeks, shorter sleep duration was predictive of greater anhedonia, reflecting the differential effects of acute versus chronic sleep deprivation in depression[56]. However, for those still of school age, longer sleep duration was associated with better sleep quality and predictive of lower depression symptom severity the following day. This was mediated by higher daytime energy level and moderated by parental enforcement of bedtimes and later school start times[57,58]. Instead, greater variation in the sleep time series data (sequence of data points occurring in successive order over a period of time) was consistently found to be associated with greater depressive symptom severity[53,59,60]. This suggests that an overall irregular sleep pattern rather than any single discreet measure of sleep may be more indicative of worsening depression, especially in young people subject to changing demands and academic pressures[55].

## Mobility

Reduced mobility was consistently found to be associated with greater depressive symptom severity in young people across a variety of different measures including step count[59], Bluetooth co-locations[54], GPS[55], time spent at home[61] and accelerometery[53,62]. No measure was predictive of later depression symptom severity at 2-year follow-up; however, this was probably the result of using a 21-day snapshot of RMT data rather than continuous monitoring for the whole 2 years (ref. 59). Indeed, again, greater variation and lower predictability in daily activity duration time series data were associated with depression symptom severity only in those participating for more than 45 days, with functioning routines becoming less stable over time as depression worsened[63].

## Smartphone use and social communication

Changes in patterns of smartphone use and social communication have been suggested as ways to infer the reduced concentration, negative cognitive biases and initial reassurance seeking[64] then subsequent social withdrawal observed in depression. Indeed, compared with healthy controls, young people with depression were found to display significantly higher smartphone use specifically in spaces across university campus dedicated to private study[53]. When using their smartphone, young people with depression also displayed significantly higher frequency of launch per app of any type[65], use of social communication apps[47,65] and daily proportion of negative words typed within social communication apps[66–68] compared with healthy controls. Lower time spent in friends' houses per visit[62] and reduced face-to-face conversation frequency and duration[54] were predictive of later depression symptom severity. Finally, as would be expected, EMA self-reported negative mood[45,60], as well as stress[54,64], self-esteem[56] and parent-reported negative mood[69] were found to be associated with, and predictive of, greater depression symptom severity.

RMT data features have been considered in isolation so far; however, their combination in linear regression and machine learning models increased predictive performance, with classification accuracy for standardized measures of depression (ability to distinguish between those with and without depression) ranging from 77% to 86% (refs. [37,53,69–71]). Further improvements were obtained through contextual filtering of RMT data features (for example, weekday/weekend sleep and daytime/evening conversation frequency)[72], normalization relative to an individual's own data[73] and inclusion of parent-report EMA responses[42,69].

**Clinical utility.** While RMT data features certainly appear relevant to depression, there is less evidence for the clinical utility of this in improving depression outcomes in young people either through objective screening, symptom management or relapse prevention. Of the 28 records pertaining to clinical utility only 10 were published studies, with the remaining 18 being registries and/or protocols of clinical trials yet to be completed[74–92].

Three studies investigated the use of RMT for one-off, opportunistic screening in young people. Comparison of RMT with face-to-face assessments of depression found fair to good concordance for the detection of positive screens of depression[35,93–97], as well as the detection of more psychosocial risk factors overall[93,95]. This suggested increased disclosure of information without fear of judgement, which may aid quicker provision of more appropriate treatment[93–97].

The use of RMT for symptom management through self-monitoring and feedback to health care professionals was often rated as useful by young people[98,99] and resulted in some reduction of depression symptom severity[100,101]. However, this reduction was not significantly different between self-monitoring and comparison groups[41,100–109], unless outliers were removed[41], or history of emotional abuse was considered[110]. As such, RMT could aid symptom management in less severe cases but may not be sufficient for those with higher depression symptom severity and/or adverse childhood experiences[41].

Many highlighted the promise of RMT for relapse prevention through detection of a 'relapse signature' and direct linkage to primary or secondary alert-based systems. RMT could even inform and directly deliver personalized digital micro-interventions that are highly focused towards the detected change requiring lower effort for purposeful engagement[111], or just-in-time adaptive interventions that adapt the provision of support to an individual's changing state over time, such that the right type of support is delivered, at the right time, just when the individual needs it most[70,71,112]. This would require understanding of which RMT data features precede relapse, their specificity for depression, their sensitivity in those already depressed, and the impact of detection on time to treat and depression outcomes, for which literature was limited, at least in young people. However, there were some small proof-of-concept studies for the prediction of day-to-day[71] or even hour-to-hour[70] fluctuations in depression symptom severity in clinical samples, leading to quick modification of care for the two patients detected and alerted as at risk[40], as well as several clinical trials underway[81,82,88,89].

## How and why?

While the overall impact of RMT on depression outcomes in young people remained inconclusive, self-monitoring specifically was consistently found to result in a greater increase in emotional self-awareness relative to comparison groups[7,99,101,103,104,113–115]. Emotional self-awareness describes the ability to understand one's own emotions and is often low is those with depression[116]; however, it can be increased through improved understanding of the links between certain events or thoughts and later emotions, behaviours and coping strategies. Indeed, this is a key component of cognitive behavioural therapy and forms the basis of many smartphone apps with EMA prompt protocols[100,113,116]. Increased emotional self-awareness can lead to improvements in other important emotional processes, such as emotional granularity (ability to specify emotions), controllability (ability to identify and control emotions to respond in a socially acceptable way) and self-regulation (ability to manage emotions)[117,118]. This may contribute to long-term behaviour change, more appropriate coping strategies and subsequent improvement in depression outcomes. Contrary to the concerns of healthcare professionals[119], self-monitoring was also found to increase readiness for and occurrence of help-seeking behaviours[102,120]. In boys specifically, RMT were found to be an alternative outlet for expressing emotion, acting to reduce the stigma they felt they would receive if they talked about their emotions until they were ready to seek help[102,120].

Where there was feedback to the health care professional, RMT facilitated quicker and more efficient communication[121], prompted recall or the start of a conversations[113] with greater disclosure of information[94], increased understanding of the young person[105], allowed shared decision-making, goal setting and tracking[7,40,98,118,122], and led to an overall increase in therapeutic engagement, alliance and adherence[75,109,113]. Finally, although limited, there was some evidence to suggest that use of RMT for depression in young people may have some unintended outcomes, which could exacerbate depression symptoms. For example, health anxiety when viewing pertinent data without a healthcare professional present to clarify what it means; occurrence of false negatives and/or positives[35,47,123]; overreliance and overcontrol, particularly in those with co-morbid health anxiety, eating disorder or obsessive–compulsive disorder where self-monitoring could cause harm[124]; burden of the data-driven nature, goal setting and behaviour change, and a sense of failure when not achieved[125]. As such, through consideration of the underlying mechanisms of what does and does not work, RMT may be better used alongside current care pathways or to monitor and/or augment the effects of other interventions for depression in young people, rather than as an intervention itself. However, the question remains as to why adherence, accuracy, predictive performance and/or effectiveness may vary across young people and different contexts.

## For whom?

Throughout the literature, there was the assumption that RMT may be a natural fit for young people who have grown up during the digital revolution. Internet access and smartphone ownership in young people were relatively high, and perspectives on the use of RMT for depression were generally positive[38,47,98,99,103,107,121,123,126–130]. However, usage statistics were quite low[127,131,132], particularly for boys[127,132] despite the benefit they may gain from RMT through reduced perceived stigma and subsequent help-seeking behaviours[102]. YPAG members indicated that the low motivation and energy experienced during depression but requirement for daily interaction over extended periods of time would/did play a large part in their decision not to use/to stop using RMT, which was reflected in the literature[27,28,133]. While no study specifically set out to investigate the impact of depression itself on the use of RMT and adherence rates as their main aim, it was still considered by some in related analyses, but with conflicting results. Five studies reported no significant effect of depression symptom severity on dropout[90,96], EMA prompt response rates[39], amount of entries/missing data[63] or exclusion due to missing data[72]. Two studies suggested that those with greater depression symptom severity may be less likely to drop out[42], with continued engagement beyond the minimum time required by the study[113]. Only one study found greater depression symptom severity, as well as treatment intensity, to be related to lower use of their smartphone app[98]. Even though the lowest adherence rates reported were comparable to those of other interventions[41,107,113], it is important to note the impact of the heterogeneous, artificially inflated or unreported quantitative measures of feasibility on the validity of adherence rates across the literature. With evidence that the level of adherence may be predictive of mental health outcomes at least in adults[134], there is a crucial need for standardization to determine whether young people

are likely to use RMT for depression, and whether this translates to real-world settings outside of the study contexts, and further investigate reasons why it may vary across individuals.

## In which contexts?

RMT were also assumed to be an increasingly ubiquitous and accessible resource that could be the much-needed scalable solution to the global mental health crisis. While there was evidence that digital divides have declined in recent years, they were still pertinent, particularly in rural areas, LMICs and/or areas with higher accessibility but low digital literacy[135,136]. As such, the widespread adoption of RMT could further perpetuate mental health inequalities and outcomes[137]. Additionally, with high initial costs in the development and implementation of RMT, LMICs sometimes resort to making use of previously established RMT that may not be suitable for the context[35]. Six studies investigated the use of RMT for depression in young people in LMICs, with a focus on adaptability to context, language and culture key to their success[35,96,136,138–140]. Qualitative results indicated that the factors influencing the type of data that should be collected, and outcome measures included to make RMT more culturally compelling and accurate in local contexts were consideration of the role of families, locally defined experience of depression, and social determinants of mental health[35,139,141,142].

There were attempts to assess the feasibility of integrating RMT into primary and secondary mental health care services; however, differing digital infrastructures and capacities made the process complicated[143,144], uptake by healthcare professionals was low due to the changes in roles, responsibilities and training required[104,113,145] with just 34% of consenting GPs actually participating in one study[104], and there was a need to balance the expectations of young people, healthcare professionals and researchers regarding alert-based systems and capacity to respond[7,35,98,108,113,119,123,129,145]. While RMT could still be used to enhance the therapeutic relationship, their intended use in the early detection of deterioration for relapse prevention is unfortunately highly unlikely to result in quicker time to treat under current resourcing and capacity levels.

Finally, the young, lived experience co-researchers highlighted the potential challenges associated with real-time monitoring in schools, where young people may not always be able to keep RMT on their person since the use of smartphones and other technologies is often not allowed during class. There were two studies investigating the use of RMT in schools[146,147], with only one including school personnel[146], whose positive perspectives were probably the result of participation bias. Some studies were able to adapt data collection methods for the school context, such as fixed-time sampling with EMA prompts occurring only outside of school time; however, data were still missed that may be crucial to understanding the role of academic stressors in depression. Further investigation of acceptability and feasibility in a greater number of schools across different regions is required, with potential to integrate RMT alert-based systems into schools instead where there is somewhat more of a capacity for early detection and relapse prevention[147]. Overall, there is a need for increased implementation science to determine whether the notable challenges above can be overcome for the sustainable integration and scaling of RMT across contexts[148].

Table 1 provides a summary of the of the ways, contexts and for whom RMT appear to work or not work for depression in young people.

## Gaps in the literature and methodological issues

This realist review has highlighted gaps where very limited evidence was available, as well as several common methodological issues that appeared across much of the literature reviewed including: convenience sampling and selection bias with those young people more interested in RMT more likely to take part; reliance on self-report in

**Table 1 | Summary of the ways, contexts and for whom RMT appear to work or not work for depression in young people**

| | |
|---|---|
| **What does and doesn't work?** | • Preference for smartphones and apps. |
| | • Passive and active monitoring, with a balance between data quality and intrusiveness. |
| | • Greater consideration of the privacy, confidentiality and ethical implications of collecting and sharing digital data needed. |
| | • Depression best detected by increased sleep variability and changes in mobility, smartphone use, social communication and self-/parent-report EMA responses. |
| | • Clinical utility in screening, self-monitoring and feedback to the healthcare professional. |
| **How and why?** | • Impact of RMT on depression outcomes unclear. |
| | Self-monitoring and feedback to healthcare professional:<br>• Increased emotional self-awareness.<br>• Enhanced therapeutic relationship.<br>• Encouraged help-seeking behaviour particularly in boys via reduced perceived stigma. |
| **For whom?** | • Potential overestimation of how much young people are likely to use RMT in real-world settings. |
| | Characteristics of those most likely to benefit from RMT for depression include:<br>• Interested in and motivated by the data-driven nature of RMT.<br>• Lower depression severity.<br>• No co-morbidities where self-monitoring could cause harm.<br>• Presence of specific behaviours or coping strategies that could be targeted to help improve depressive symptoms. |
| **In which contexts?** | • Monitoring during transition to university, known to be associated with worsening depression. |
| | • Notable challenges with the integration of alert-based systems into primary and secondary healthcare likely to limit the potential of RMT in relapse prevention. |
| | • Schools may not allow RMT during class where a large proportion of a young person's time is spent but important data missed. |
| | • Potential perpetuation of mental health inequalities without consideration digital divides and adaptability to context, language and culture. |

community samples with few reporting the number above/below clinical cut-offs; potentially inappropriate statistical analyses for the low sample sizes but large time series datasets[149]; short study duration (7 days to 12 months at most) or snapshot monitoring periods (2–3 weeks) in multiyear studies despite estimated relapse rates to be only 5% within the first 6 months, 12% by 12 months, 40% by 2 years, then 70% by 5 years (refs. 150–152); low number of predictor variables and little consideration of the many factors that could have confounding, mediating or moderating effects over the length of the monitoring period; potentially conflicting indicators of depression symptom severity due to the diversity of experience overlooked[72,73,123]; heterogeneity, artificial inflation, or non-report of quantitative measures of feasibility making comparison across studies difficult, especially with regard to adherence[32,39]; limited investigation of specificity, sensitivity and impact on time to treat and depression outcomes; and overall little attempt to assess and explain variability in adherence, accuracy and predictive performance across individuals[70].

## Realist intervention theory

Evidence synthesis allowed gradual refinement of the initial framework (Fig. 2) into a realist intervention theory comprising context–mechanism–outcome (CMO) configurations explaining patterns of use of RMT for depression in young people (Fig. 3).

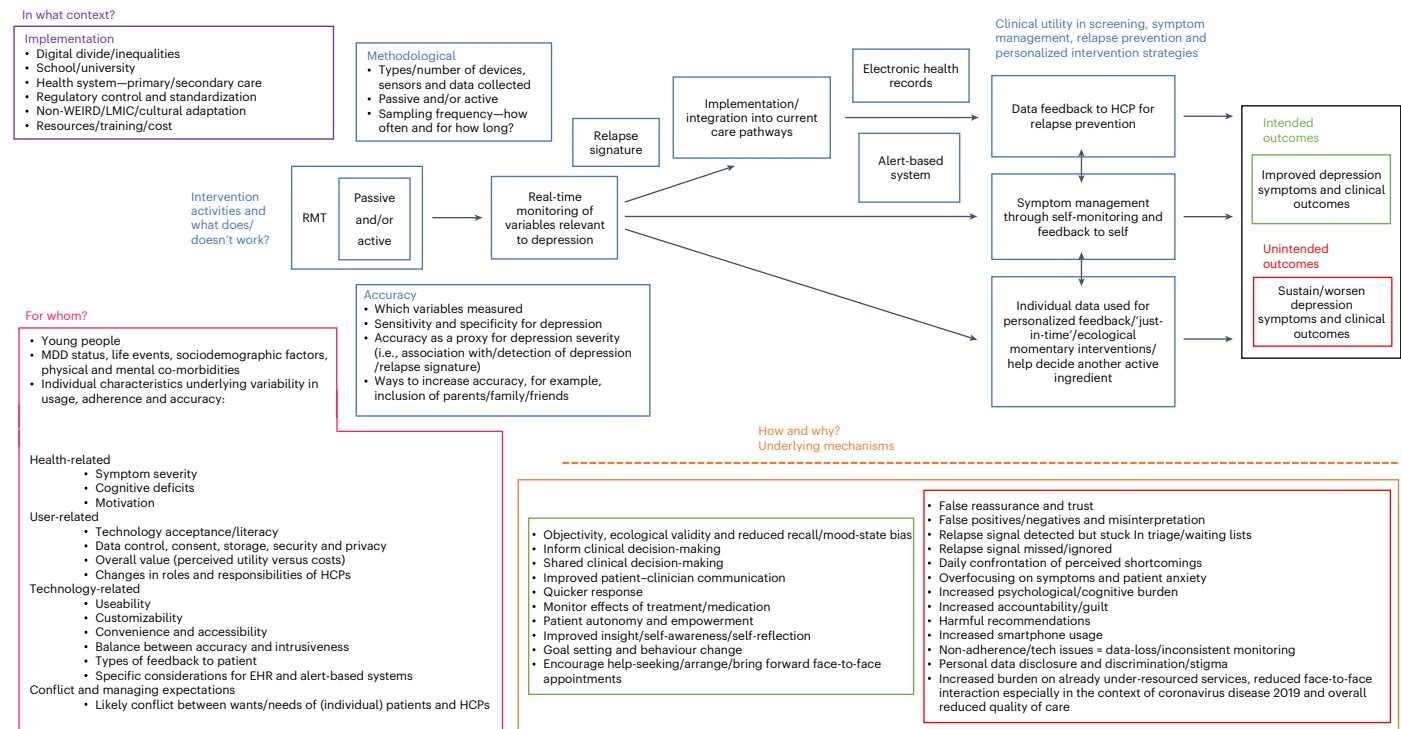

**Fig. 2 | A theoretically based evaluative framework to be populated with evidence for the ways, contexts and for whom RMT appear to work or not work for depression in young people.** Blue text boxes list factors potentially influencing the ways in which RMT may work. Pink text boxes list factors potentially influencing who RMT may work for. The purple text box lists factors potentially influencing the contexts in which RMT may work. The orange text box lists potential mechanisms (M) linking these influencing factors (C) to outcomes (O), with those linked to intended outcomes in green, and those linked to unintended outcomes in red. RMT, remote measurement technologies; MDD, major depressive disorder; HCPs, healthcare practitioners; WEIRD, Western, educated, industrialised, rich and democratic; LMIC, low-middle income country.

## Discussion

There has been an extensive movement towards digital mental health in recent years, with RMT suggested as the scalable solution to the global mental health crisis[22]. However, in the wake of this steep growth, regulation and standardization remain inadequate, with limited understanding of the effect on depression outcomes nor scrutiny of the realist and ethical considerations likely to greatly impact the benefits, implementation and overall potential of RMT in the real world[23–26]. Therefore, a realist review was conducted in collaboration with The McPin Foundation YPAG through which a refined, realist intervention theory was produced explaining the ways, for whom, in which contexts, how and why RMT appear to work or not work for depression in young people aged 14–24 years.

Smartphones were most preferred, with both passive and active data collection for a holistic approach but a balance between data quality, intrusiveness and data privacy. From the evidence currently available, depression was best detected by changes in sleep, mobility, smartphone use, social communication and self- or parent-reported mood. This had some uses in screening, self-monitoring and feedback to the healthcare professional but not in relapse prevention and personalized interventions, where more research is required. The impact of RMT as an intervention itself on depression outcomes remained unclear, but self-monitoring and feedback improved emotional self-awareness, therapeutic relationship and help-seeking behaviours. With limited standardization and investigation of the impact of depression on adherence rates, there may be an overestimation of how much young people are likely to use RMT in the real world. However, they were most likely to benefit those interested in and motivated by the data-driven nature, who have lower depression severity, no co-morbidities where self-monitoring could cause harm, and the presence of changeable behaviours. RMT facilitated monitoring during transition to university, known to be associated with worsening depression in young people; however, there were significant challenges in health care and school settings. Adaptability was important, such that RMT were culturally compelling and accurate for the local context. The realist review also highlighted gaps where very limited evidence was available, as well as several common methodological issues that appeared across much of the literature. With the evidence base as it stands and important insights from those with lived experience, we make the following recommendations for the use of RMT for depression in young people.

Recommendations for future research: before moving forward, ethical procedures for the collection, sharing and use of digital data need to be reviewed and updated, a standard set of quantitative measures of feasibility and depression outcome measures used in digital mental health research need to be decided, and other methodological issues need to be rectified to increase the validity of results. Future research should then focus on (1) potential unintended outcomes of real-time monitoring; (2) predictive investigations to determine which, if any, RMT data features precede and have sensitivity and specificity for depression relapse; (3) use of RMT to inform and deliver digital micro-interventions and just-in-time adaptive interventions; (4) impact of the use of RMT on time to treat and depression outcomes in young people; and (5) implementation science to determine whether the notable challenges can be overcome.

Recommendations for practice: without the above in place, the current best use of RMT for depression in young people is for self-monitoring and feedback to the healthcare professional where possible, to increase emotional self-awareness, enhance the therapeutic relationship and monitor the effectiveness of other interventions. There was resounded consensus from YPAG members that this could become part of a blended, stepped-care approach, with use throughout

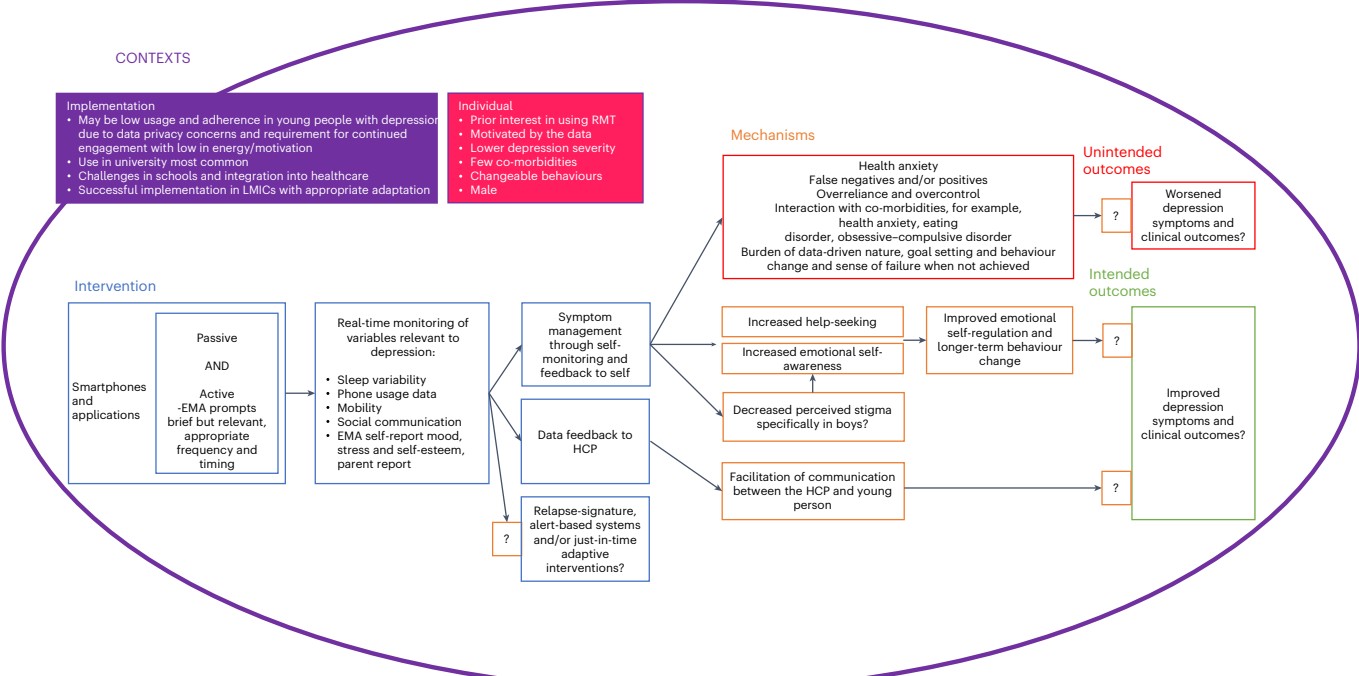

**Fig. 3 | Realist intervention theory of RMT for depression in young people.** Blue text boxes list factors found to influence the ways in which RMT may work. The pink text box lists factors found to influence who RMT may work for. The purple text box lists factors found to influence the contexts in which RMT may work. The orange text boxes list mechanisms (M) found to link these influencing factors (C) to intended (green) and unintended (red) outcomes (O). Question marks indicate areas where evidence is lacking and there is need for further research. RMT, remote measurement technologies; EMA, ecological momentary assessment; HCPs, healthcare practitioners; LMIC, low-middle income country.

the care pathway from opportunistic screening, watchful waiting, triage and placement on waiting lists, to during intervention; however, the impact of this on depression outcomes remains to be tested.

Limitations: the realist review allows inclusion of a range of literature, actively encourages stakeholder consultation and can provide greater understanding as to why an intervention may or may not work, when used by different individuals, in different contexts to help decision-makers implement the intervention most effectively. However, the realist intervention theory and recommendations are only as strong as the literature included in the review, and with limits on how much literature one can cover, the quality of the range of literature included, and the absence of formal quality appraisals or privileging of certain study designs[153], no definitive conclusions about whether an intervention will work or not can be made.

## Methods
### Rationale for the realist approach
While traditional methods of review focus on determining the overall effectiveness of an intervention, a realist review seeks to determine the ways in which an intervention does or does not work, when used by different individuals, in different contexts, and why, then makes recommendations for how the intervention can be implemented and used most effectively[153]. It is an iterative process encouraging consultation with key stakeholders throughout[153], who by virtue of their lived experience bring a wealth of additional expertise, knowledge and insights important for real-world impact[154]. In the present study, the realist review was conducted in accordance with the Realist and Meta-narrative Evidence Syntheses: Evolving Standards (RAMESES) for quality and publication[155] (see Supplementary Table 1—RAMESES Checklist), and in collaboration with The McPin Foundation YPAG.

### Collaboration with McPin YPAG
The McPin YPAG comprises 20 members aged 13–28 years from across the United Kingdom, all with lived experience of different mental health issues. Two 2-h consultations were held with 13 members of the McPin YPAG, one at the start to clarify scope and help develop the initial list of intervention theories for the use of RMT for depression in young people, and one near the end to gather final evidence to support, refute or further refine the intervention theory. Additionally, two young people co-researchers (G.N. and T.S.) with relevant lived experience were recruited from the YPAG by emailing a lay summary of the project, person specification and a request for expressions of interest. The two young people co-researchers were trained in realist review methodology over three 2-h sessions by A.E.L.W., after which they were involved in shortlisting key intervention theories, deciding search terms and inclusion criteria, refining them in light of emerging evidence, literature selection and appraisal, data extraction, evidence synthesis and developing the recommendations. An example of how this realist approach and collaboration with young people influenced the process was suggestion to include the following in the initial framework: influence of school rules, motivation for continued engagement, and views on data privacy (for specific detail on how the perspectives of the two young, lived experience co-researchers and wider YPAG members shaped the study, see Supplementary Table 2—Completing the Feedback Circle).

### Clarifying scope
Exploratory searches were conducted in Google Scholar and PubMed using just the three search terms, 'remote measurement technologies', 'depression' and 'young people', then snowball sampling, with broad selection criteria, including any literature on the use of RMT in mental health. Through discussion with the YPAG and co-researchers and progressive focusing of scope, a final list of relevant intervention

theories was produced. These were then categorised to form a theoretically based evaluative framework to be populated with evidence (Fig. 2). The framework also formed the basis of the search strategy and bespoke data extraction template, with inputs from the co-researchers on sources of literature, search terms, refinement of selection criteria and data to be extracted.

## Search processes

Searches were conducted across ten electronic databases, including those with a more specialist focus on health technology (PubMed; Ovid—EMBASE, MEDLINE, APA PsychINFO and Global Health; Web of Science; Cochrane Library; IEEE Xplore; HTA database; and ACM digital library) and seven sources of grey literature (CADTH, NICE, WHO, ClinicalTrails.gov, ISRCTN registry, Gov.uk and rxiv.org), with a date restriction of January 1990 to current day (August 2021) reflecting the digital revolution. Searches included various terms for RMT, depression, young people and those relating to the specific intervention theories included in the framework. Of note was the varying terminology used for RMT due to its novelty and current lack of nomenclature standardization in the field making searches complex; however, we have tried to be as comprehensive as possible (see Supplementary Table 3—Searches for the search strings used in each database). Iterative searches involved adding in specific search terms for further testing and refinement of intervention theories, or where evidence was sparse. Hand searches of reference lists of eligible literature and relevant systematic reviews were also conducted. Search results were downloaded for import into an external reference management software (EndNote) and automatic deduplication, before export into excel for literature selection, appraisal and data extraction.

## Literature selection criteria and appraisal

A realist review would normally include literature from other fields of research to aid interpretation then selection criteria refined throughout the realist review process in light of the emerging data[30]. Indeed, the co-researchers were keen to include literature on bipolar disorder, which makes up a large volume of the evidence base for the use of RMT in depressive episodes and mental health[20]. However, given the focus of the review and time constraints, more stringent selection criteria than usual were set from the start then slightly refined. Initial selection criteria included: (1) peer-reviewed articles and grey literature; (2) of any type including reviews, quantitative, qualitative and mixed-methods original research and pilot/acceptability/feasibility/efficacy clinical trials, so long as the methodology used to generate the extracted data was credible (rigour); (4) discussing the use of RMT in depression in young people (relevance); and (5) clinical or community populations allowing inclusion of subclinical but at-risk youth who could deteriorate during the monitoring period. Literature considering depression alongside its common mental and physical co-morbidities was still included as this will be an important factor when assessing who RMT will and won't work for. Literature was excluded if (1) the technology was used solely for providing treatment (for example, computerized-cognitive behavioural therapy) without any remote measurement; (2) only a specific type of depression was considered (for example, bipolar, perinatal or postnatal), depression was not the primary condition, or no standardized measure of depression was included (for example, wellbeing scales); and (3) it specifically stated the sample was older adults or the elderly. Refinements made to the selection criteria were (1) a specific focus on studies where the mean age of young people fell within the 14–24 years age range as defined by the WHO and United Nations[156,157], unless the sample was considering other end users of RMT involved in the care of young people (for example, healthcare professionals or parents); (2) exclusion of all literature where the main use of the technology was providing treatment, even if there was some capacity for remote measurement as it became very difficult to disentangle the distinct effects of each; and (3) exclusion of systematic reviews once reference lists had been hand-searched and

it had been determined that all relevant literature included in the review had already been included in the present study.

## Data extraction and evidence synthesis

The initial framework was used as a basis for the development of a bespoke literature selection, appraisal and data extraction template in excel, with columns for the input of extracted data relating to contexts, mechanisms and outcomes, and each record on a new row (see Supplementary Table 4—Literature Selection, Appraisal and Data Extraction Form). The same template was used for data extraction from all literature, as, while a particular record may focus on a particular intervention theory, there may still be data present that could provide evidence or refinement of another intervention theory. Records were classified and sorted by intervention theory, and then divided between the team lead (A.E.L.W.), co-researchers (G.N. and T.S.) based on their areas of interest, and two postdoctoral researchers (Z.Z. and M.M.) for data extraction. Initial evidence synthesis was undertaken by the team lead, with inferences then discussed in a consultation with the YPAG to ensure external validity against the lived experience of young people, and final review by the co-researchers to ensure perspectives were interpreted correctly without researcher bias. As part of the process of evidence synthesis, judgements of relevance and rigour of the evidence were made. Relevance refers to whether evidence from the literature or consultations is relevant to intervention theory development. Rigour refers to whether the evidence from the literature or consultations is sufficiently trustworthy to be used to refine the intervention theory. Some sources of evidence were privileged over others, such as peer-reviewed literature or when there was consensus across all 13 members of the McPin YPAG; however, realist review generally rejects the use of a privileging/hierarchical approach to study design, seeing merit in a wide range of evidence from diverse sources. Additionally, study design is an inappropriate aspect to focus on when in a realist review, only a single element of the study may be considered when identifying, testing and refining a specific theory[153]. Where further testing and refinement of intervention theories was required, these literature search, selection, appraisal, extraction and synthesis processes were repeated. In this way, the initial framework was gradually focused into a refined, realist intervention theory comprising CMO configurations describing the ways, for whom, in which contexts and why RMT appear to work or not work for depression in young people.

## Reporting summary

Further information on research design is available in the Nature Portfolio Reporting Summary linked to this article.

# Data availability

This manuscript comprises a realist review, for which a variety of databases were searched for relevant literature, from which data were extracted. The databases used are listed below, and all included literature has been referenced, but data availability may be limited depending on whether or not the database/literature is open access. Web links are provided for publicly available databases. PubMed (https://pubmed.ncbi.nlm.nih.gov/); Ovid (EMBASE, MEDLINE, APA PsychINFO and Global Health); Web of Science; Cochrane Library (https://www.cochranelibrary.com/); IEEE Xplore (https://ieeexplore.ieee.org/Xplore/home.jsp); HTA database (https://database.inahta.org/); ACM digital library (https://dl.acm.org/); CADTH (https://www.cadth.ca/); NICE (https://www.nice.org.uk/); WHO (https://www.who.int/); ClinicalTrails.gov (https://clinicaltrials.gov/); ISRCTN registry (https://www.isrctn.com/); Gov.uk (https://www.gov.uk/); arXiv (https://arxiv.org/).

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

## Acknowledgements

This work was funded by the Wellcome Trust Mental Health 'Active Ingredients' 2021 commission awarded to A.E.L.W. at King's College London. V.M. and A.E.L.W. are also supported by a MQ Brighter Futures grant (MQBF/1 IDEA), and V.M. by the National Institute for Health Research Biomedical Research Centre at South London and Maudsley NHS Foundation Trust and King's College London (NIHR Maudsley BRC). The funders had no role in study design, data collection and analysis, decision to publish or preparation of the manuscript. The views expressed are those of the authors and not necessarily those of the Wellcome Trust, MQ, NHS, NIHR, Department of Health and Social Care, or King's College London. We thank The McPin Foundation and their YPAG for their substantial contributions to the project, manuscript and dissemination.

## Author contributions

A.E.L.W., A.v.H. and V.M. conceptualized the project. A.E.L.W., G.N. and T.S. conducted the project work, with Z.Z. and M.M. contributing to data extraction.

## Competing interests

A.E.L.W. has received funds from The McPin Foundation in her role as a lived experience panel member on an unrelated project, and is now employed by the McPin Foundation as a Public Involvement in Research Manager. G.N. is now employed by the McPin Foundation as a Peer Researcher. A.v.H. developed the Electronic Behaviour Monitoring app (EBM version 2.0) as part of the StandStrong platform for passive sensing in maternal depression in low resource settings. T.S., M.M., Z.Z. and V.M. declare no competing interests.

## Additional information

**Correspondence and requests for materials** should be addressed to Annabel E. L. Walsh.

# Reporting Summary

## Statistics

For all statistical analyses, confirm that the following items are present in the figure legend, table legend, main text, or Methods section.

| n/a | Confirmed | |
|---|---|---|
| ☐ | ☒ | The exact sample size (*n*) for each experimental group/condition, given as a discrete number and unit of measurement |
| ☒ | ☐ | A statement on whether measurements were taken from distinct samples or whether the same sample was measured repeatedly |
| ☒ | ☐ | The statistical test(s) used AND whether they are one- or two-sided *Only common tests should be described solely by name; describe more complex techniques in the Methods section.* |
| ☒ | ☐ | A description of all covariates tested |
| ☒ | ☐ | A description of any assumptions or corrections, such as tests of normality and adjustment for multiple comparisons |
| ☐ | ☒ | A full description of the statistical parameters including central tendency (e.g. means) or other basic estimates (e.g. regression coefficient) AND variation (e.g. standard deviation) or associated estimates of uncertainty (e.g. confidence intervals) |
| ☒ | ☐ | For null hypothesis testing, the test statistic (e.g. *F*, *t*, *r*) with confidence intervals, effect sizes, degrees of freedom and *P* value noted *Give P values as exact values whenever suitable.* |
| ☒ | ☐ | For Bayesian analysis, information on the choice of priors and Markov chain Monte Carlo settings |
| ☒ | ☐ | For hierarchical and complex designs, identification of the appropriate level for tests and full reporting of outcomes |
| ☒ | ☐ | Estimates of effect sizes (e.g. Cohen's *d*, Pearson's *r*), indicating how they were calculated |

*Our web collection on statistics for biologists contains articles on many of the points above.*

## Software and code

Policy information about availability of computer code

| Data collection | Clarivate EndNote 20 Microsoft Office 365 Excel |
|---|---|
| Data analysis | Microsoft Office 365 Excel |

For manuscripts utilizing custom algorithms or software that are central to the research but not yet described in published literature, software must be made available to editors and reviewers. We strongly encourage code deposition in a community repository (e.g. GitHub). See the Nature Portfolio guidelines for submitting code & software for further information.

## Data

Policy information about availability of data

All manuscripts must include a data availability statement. This statement should provide the following information, where applicable:

- Accession codes, unique identifiers, or web links for publicly available datasets
- A description of any restrictions on data availability
- For clinical datasets or third party data, please ensure that the statement adheres to our policy

This manuscript comprises a realist review, for which a variety of databases were searched for relevant literature, from which data was extracted. The databases used are listed below, and all included literature has been referenced, but data availability may be limited depending on whether or not the database/literature is open access. Web links are provided for publicly available databases.

PubMed – https://pubmed.ncbi.nlm.nih.gov/; Ovid (EMBASE, MEDLINE, APA PsychINFO and Global Health); Web of Science; Cochrane Library – https://www.cochranelibrary.com/; IEEE Xplore – https://ieeexplore.ieee.org/Xplore/home.jsp; HTA database – https://database.inahta.org/; ACM digital library – https://dl.acm.org/; CADTH – https://www.cadth.ca/; NICE – https://www.nice.org.uk/; WHO - https://www.who.int/; ClinicalTrails.gov – https://clinicaltrials.gov/; ISRCTN registry – https://www.isrctn.com/; Gov.uk – https://www.gov.uk/; arXiv – https://arxiv.org/.

# Research involving human participants, their data, or biological material

Policy information about studies with underline{human participants or human data}. See also policy information about underline{sex, gender (identity/presentation), and sexual orientation} and underline{race, ethnicity and racism}.

| | |
|---|---|
| Reporting on sex and gender | N/A |
| Reporting on race, ethnicity, or other socially relevant groupings | N/A |
| Population characteristics | Not research participants, rather lived experience involvement via consultation with the McPin Young People's Advisory Group (YPAG) (14 members, aged 14 - 25), and 2 young people co-researchers (1 male, 1 female) with lived experience of past history and/or current depression. Whilst demographic data is considered upon recruitment to the YPAG/involvement opportunities to ensure diversity of voices, we do not store this data and therefore cannot include it here. |
| Recruitment | Recruited through The McPin Foundation |
| Ethics oversight | The McPin Foundation |

Note that full information on the approval of the study protocol must also be provided in the manuscript.

# Field-specific reporting

Please select the one below that is the best fit for your research. If you are not sure, read the appropriate sections before making your selection.

☐ Life sciences   ☒ Behavioural & social sciences   ☐ Ecological, evolutionary & environmental sciences

For a reference copy of the document with all sections, see nature.com/documents/nr-reporting-summary-flat.pdf

# Behavioural & social sciences study design

All studies must disclose on these points even when the disclosure is negative.

| | |
|---|---|
| Study description | Realist review of qualitative, quantitative, and mixed-methods study designs, as well as conference proceedings, protocols, and other gray literature. |
| Research sample | Previously published literature relevant to the use of remote measurement technologies for depression in young people aged 14 - 24 years. |
| Sampling strategy | Exploratory searches, followed by iterative purposive searches, as well as snowball sampling and hand searching references. Searches were as comprehensive as possible, but the amount of evidence available for synthesis limited by the amount of relevant previously published literature available. |
| Data collection | A bespoke data extraction form was created in Excel by AW, with evidence extracted from relevant previously published literature and inputted into the form by AW, GN, TS, ZZ & MM. No researchers were blind to the study hypothesis. |
| Timing | Searchers were conducted in August 2021. |
| Data exclusions | Literature excluded n = 5703. Literature where remote measurement technologies were used solely to deliver an intervention without any remote measurement, focused on a specific type of depression (e.g., bipolar, perinatal, or postnatal), or did not include a standardized measure of depression (e.g., well-being scales) was excluded. |
| Non-participation | Not applicable - the study was a realist review of relevant previously published literature, with no human participants involved in the study. |
| Randomization | Not applicable - the study was a realist review of relevant previously published literature, with no human participants involved in the study. |

# Reporting for specific materials, systems and methods

We require information from authors about some types of materials, experimental systems and methods used in many studies. Here, indicate whether each material, system or method listed is relevant to your study. If you are not sure if a list item applies to your research, read the appropriate section before selecting a response.

## Materials & experimental systems

| n/a | Involved in the study |
|-----|----------------------|
| ☒ ☐ | Antibodies |
| ☒ ☐ | Eukaryotic cell lines |
| ☒ ☐ | Palaeontology and archaeology |
| ☒ ☐ | Animals and other organisms |
| ☒ ☐ | Clinical data |
| ☒ ☐ | Dual use research of concern |
| ☒ ☐ | Plants |

## Methods

| n/a | Involved in the study |
|-----|----------------------|
| ☒ ☐ | ChIP-seq |
| ☒ ☐ | Flow cytometry |
| ☒ ☐ | MRI-based neuroimaging |

