## [Peer Review File · Nature Human Behaviour]

Peer Review Information

Journal: Nature Human Behaviour

Manuscript Title: A collaborative realist review of remote measurement technologies for depression in young people

Corresponding author name(s): Annabel E L Walsh

Reviewer Comments & Decisions:

Decision Letter, initial version:
--

14th July 2022

Dear Dr Walsh,

Thank you once again for your manuscript, entitled "A realist review and lived experience involvement approach to remote measurement technologies for depression in young people.," and for your patience during the peer review process.

Your manuscript has now been evaluated by 3 reviewers, whose comments are included at the end of this letter. Although the reviewers find your work to be of interest, they also raise some important concerns. We are [very] interested in the possibility of publishing your study in Nature Human Behaviour, but would like to consider your response to these concerns in the form of a revised manuscript before we make a decision on publication.

To guide the scope of the revisions, the editors discuss the referee reports in detail within the team, including with the chief editor, with a view to (1) identifying key priorities that should be addressed in revision and (2) overruling referee requests that are deemed beyond the scope of the current study. We hope that you will find the prioritised set of referee points to be useful when revising your study. Please do not hesitate to get in touch if you would like to discuss these issues further.

1) All three reviewers request greater transparency of reporting regarding your Methods. We ask that you address these concerns in full. To facilitate this, please include in your resubmission a completed copy of the RAMESSES checklist (<https://bmcmedicine.biomedcentral.com/articles/10.1186/1741-7015-11-21/tables/1>), and ensure that all items are fully and transparently reported in your manuscript. This should include the addition of a flow diagram in your Results (<https://bmcmedicine.biomedcentral.com/articles/10.1186/1741-7015-11-21#Fig1>). Additionally, you may want to follow Reviewer 1's recommendation to provide a PRISMA checklist.

2) Relatedly, please add a Table of all included studies, as suggested by reviewers 1 and 2.

3) All reviewers raise the common concern that your conclusions as they are currently written are not justified by the current state of the literature. Please modify your conclusions accordingly, and ensure that you have a clear rationale for the conclusions you draw from your review.

In sum, we invite you to revise your manuscript taking into account all reviewer and editor comments. We are committed to providing a fair and constructive peer-review process. Do not hesitate to contact us if there are specific requests from the reviewers that you believe are technically impossible or unlikely to yield a meaningful outcome.

We hope to receive your revised manuscript within two months. I would be grateful if you could contact us as soon as possible if you foresee difficulties with meeting this target resubmission date.

- Include a "Response to the editors and reviewers" document detailing, point-by-point, how you addressed each editor and referee comment. If no action was taken to address a point, you must provide a compelling argument. When formatting this document, please respond to each reviewer comment individually, including the full text of the reviewer comment verbatim followed by your response to the individual point. This response will be used by the editors to evaluate your revision and sent back to the reviewers along with the revised manuscript.
- Highlight all changes made to your manuscript or provide us with a version that tracks changes.

[REDACTED]

We look forward to seeing the revised manuscript and thank you for the opportunity to review your work. Please do not hesitate to contact me if you have any questions or would like to discuss these revisions further.

Sincerely,

Charlotte Payne

Charlotte Payne, PhD
Senior Editor
Nature Human Behaviour

Reviewer expertise:

Reviewer #1: mHealth, adolescents

Reviewer #2: youth mental health, evidence syntheses

Reviewer #3: realist reviews, digital health

REVIEWER COMMENTS:

Reviewer #1:

Remarks to the Author:

It was a pleasure reading this review which covered a large and rapidly growing literature investigating the use of remote measurement technologies (RMT) for depression in young people. The authors conducted a realist review of 104 studies and included co-researchers with lived experience in their group. Broadly, they concluded that RMT are capable of detecting depression while the evidence that RMT effectively reduce depression or are acceptable was unclear. The authors acknowledge that many parts of the literature contain important gaps (e.g., unintended consequences, application to low and middle income countries, evidence for efficacy).

I think this study has many strengths. It was well written. The authors covered a large literature with a nice balance between sufficient detail and helpful summaries. Their figures were generally helpful (although I found Figure 2 more difficult to follow). I really appreciated their intention of including lived experience co-researchers in the study. However, I do have some suggestions that I think are worth addressing:

1. I found the description of the methods a bit opaque (e.g., lines 163 to 176). Most notably, it was not clear to me how the authors actually synthesized the literature. I think this is very important to clarify, as it seems a different research group could theoretically read the same literature and draw divergent conclusions. The review "felt" fairly balanced to me, but it would be very helpful to have details about how this synthesis occurred. I was particularly curious how the authors handled conflicting findings and null results. While these were discussed in some places, I am guessing there were many more instances that could not be fully represented. In addition, I think it would be worth including a table with information that was coded for each study.

2. Relatedly, it was also not clear to me how precisely the co-researchers' perspectives impacted the study - what was changed / informed as a result of their inclusion? I appreciate the importance of including co-researchers with lived experience. However, lacking some specifics around this process and its impact, it can seem a bit like it was done for show rather than actually informing the final product. (I am not suggesting that is the case, but simply noting this potential impression.) I did appreciate the several places in the text where the co-researchers' perspective was noted specifically. On the other hand, I do have some questions about drawing conclusions / overweighting the perspective of two individuals. It would be helpful to be more specific about which opinions attributed to YPN were representative of the broader group vs. the co-researchers alone.

3. Also related to methods, I was curious how the authors chose which evidence to privilege. From my

perspective, it would make sense to privilege meta-analytic / systematic review evidence, following by randomized trials, etc. Did the authors apply some evaluation of the rigor of the reviewed studies in deciding which studies to discuss more?

4. One broad question that arose for me is whether it is reasonable to think about RMT as an intervention. It seems that much of the available literature at this time is simply trying to determine whether depression can be assessed using RMT. Of course, eventually the goal is to have these tools translated into interventions, but is it reasonable to evaluate RMT as an intervention at this stage? From the perspective that RMT is not necessarily intended to be an intervention, some statements (e.g., RMT as "much-needed scalable solution to the global mental health crisis", Line 383) seem a bit unreasonable.

5. The authors describe Fig 2 as a "realist intervention theory." This may be primarily semantic, but I have some trouble seeing how this figure is a theory, per se. It seems to me more a visual representation / summary of the current state of the evidence with some theoretical pathways highlighted. I would suggest using different terms to describe this figure and not referencing a theory in the abstract.

6. While much of the discussion was nuanced, some statements seemed overly broad. For example, Line 230 states that "RMT data features were able to distinguish those with depression from healthy controls." However, the authors then go on to discuss several instances in which findings were null or inconsistent (e.g., various sleep parameters). I wonder if the general statements like that on Line 230 could be reconsidered for accuracy.

7. The authors conclude that the best use of RMT is within a blended, stepped-care approach (Line 468-470). However, it was not clear to me how precisely they arrived at this conclusion. It seems to me it may be premature to make recommendations for practice based on the state of the literature. Did they review studies that suggested RMT is useful within a blended, stepped-care approach? And, if not, what supports them making this conclusion?

Minor suggestions:

1. Even though the authors did not conduct a meta-analysis, I think following the PRISMA guidelines would be helpful (e.g., including exact search terms, completing the PRISMA checklist, etc.).

2. I was not sure what "worry in the absence of safeguarding" (Line 344) meant.

3. Some parts of Figure 2 were not clear to me. For example, what did the arrow between active and passive mean? Were all the bullet points in the "Usage & Adherence" box related to lower usage and adherence? What did the bullet points in the "Accuracy" box mean - are these features that improve accuracy? A clear table note would help improve the interpretability, I think.

4. On Line 460, the authors argue that depression outcomes need to be standardized. Don't we have widely available standardized measures of depression?

Reviewer #2:

Remarks to the Author:

Key results: Please summarise what you consider to be the outstanding features of the work.

The main finding of this review is not so much about what we do know, but more so what we do not

yet know about the safety and efficacy of remote measurement technologies (RMTs) for youth depression. Individually, the findings are of moderate value (e.g., identifying measurements that best predict depression, utility of RMTs, some understanding of how and why RMTs work, some suggestions for whom they work), but the overarching implication of these findings is the outstanding feature of this work: that the enthusiasm for RMTs to identify and treat youth depression needs to be met with an equal enthusiasm for high quality research and practice that takes into account data privacy, unintended outcomes of RMTs, and clinical utility for a broader range of contexts, including those where access to RMTs is more limited. This is an important review and I hope that the following comments and suggestions are useful to the authors.

Validity: Does the manuscript have flaws which should prohibit its publication? If so, please provide details.

No.

Originality and significance: What are the major claims of the paper? Do you think that they represent a significant advance in the field? If the conclusions are not original, please provide relevant references. On a more subjective note, do you feel that the results presented are of immediate interest to many people in your own discipline, and/or to people from several disciplines?

Although there have been several reviews in the area, including recently published reviews, to my knowledge this is the first to undertake a realist review approach and to have such significant lived experience involvement. Some findings are consistent with previous reviews; however, this review represents an advance in our understanding of the role that context plays in the effectiveness of RMTs for youth depression. The integrated finding regarding concerns over data privacy are novel and are timely given the simultaneous rise in data security issues and increasing push towards digital solutions for youth mental health.

Data & methodology: Please comment on the validity of the approach, quality of the data and quality of presentation. Please note that we expect our reviewers to review all data, including any extended data and supplementary information. Is the reporting of data and methodology sufficiently detailed and transparent to enable reproducing the results?

Overall, the quality of presentation was high and the rationale for conducting a realist review in this area was strong.

There were no data files presented. I expected to see a number of supplementary files that I did not (I have made note of these throughout my comments).

Although the reporting included descriptions of items listed on the Realist and Meta-narrative Evidence Syntheses: Evolving Standards (RAMESES) for quality and publication, at times I felt as if I was reading a statement that you had done it, but not enough to show me how you did it or what impact it had on the development/refinement of the theory. A table of the RAMESES items with notes about how the manuscript addressed each item would have been a useful supplementary file.

I have included additional areas that I felt were lacking in the 'suggested improvements' section.

Preregistration: If any part of the work reported in the manuscript was pre-registered, did the authors follow their preregistration plan? Did they report any deviations from their preregistration? Note that we ask authors to provide a link to the pre-registration in the Methods section and state the date of pre-registration. We also ask that authors disclose all deviations from the pre-registered protocol and

explain the rationale for deviation (e.g., flaw, suboptimality, or reviewer/editorial request). In cases of deviation from the analysis plan, the originally planned analyses need to be reported in Supplementary Information.

N/A (I cannot see any information about preregistration).

Appropriate use of statistics and treatment of uncertainties: Please include in your report a specific comment on the appropriateness of any statistical tests, and the accuracy of the description of any error bars and probability values.

N/A

Custom code: If the work includes custom code, does the code run as intended? If you are unable to access the code, please contact us.

N/A

Conclusions: Do you find that the conclusions and data interpretation are robust, valid and reliable?

Overall, I felt that the conclusions were robust in terms of the chosen methodology (i.e., meeting the requirements of a realist review). There were a number of areas where I felt additional details could either be included or excluded, and these are listed in the 'suggested improvements' section.

Suggested improvements: Please list additional analyses, experiments or data that could help strengthening the work in a revision.

Key elements that were missing for me:

- I expected to see you state the theories more clearly and to show how they were refined by the end of the review and what influenced those refinements.
- A table of included studies that provided information about the study design, geographic location/s, setting, age/gender/ethnicity of participants, diagnostic information where available, etc.

Whole manuscript:

- Minor point: please check manuscript for long sentences that impede readability and could be broken into two sentences (there are some in each section, but the 'sleep' section on page 8 is a good example, as is the discussion of unintended consequences on pages 10-11).
- Optional suggestion: If you anticipate broad readership and you wish for the manuscript to be more understandable (e.g., by young people), you could consider including brief definitions for some of the technical jargon and phrases (e.g., sleep latency, time series data, accelerometry, classification accuracy, differences between emotional granularity, controllability, and self-regulation).

Introduction:

- Some claims were missing a reference.
- The statement that diagnosing depression "can be challenging with a lack of objective, reliable diagnostic criteria..." feels out of place with the rest of the review (e.g., arguing for standardised measures of depression, referring to RMT measures as objective). The contradiction left me wondering what assumptions the authors have about the validity and reliability of depression diagnosis, and how these views potentially influenced the review. After you made the point about lack of reliable diagnostic criteria, I anticipated an interesting discussion about the tension of using RMTs to identify or predict something (i.e., diagnosis) that is associated with such challenges. When this didn't happen, I wondered if you instead meant that the criteria were reliable, but measures to assess whether

someone meets these criteria are unreliable.

- Minor point: given their recent emergence, it would be good to define 'just-in-time adaptive interventions' for readers in the first instance.
- Page 3, line 108: you note that few clinical trials have been conducted, yet earlier you refer to this as a rapidly growing area of research. It may leave the reader wondering what type of research has been done if they are unfamiliar with the area. Including a sentence here would be useful. In the same sentence, it would be good to clarify if you are talking about effectiveness of RMTs in identifying or treating depression (or both).

Methods:

- Lived experience involvement is a strength of this review. It would be good to describe in general terms if lived experience involvement included people who had experienced use of RMTs for depression.
- Inclusion criteria: can you please be more specific about the age range. Were studies included if the age of all participants was in the 12-24 age range? Or if the mean age was within the 12-24 age range?
- Exclusion criteria: please include brief justification of excluding specific types of depression.
- Data extraction and evidence synthesis: please see comment above (it would be really useful if you could include a figure or supplementary file of the initial intervention theories, how they changed over time, and what influenced these changes).

Results:

- Minor point: page 7, line 225, consider changing 'is' to 'are'.
- Minor point: on page 7, line 230, do you mean to say 'some RMT data features'?
- Minor point: some signposting is needed at the end of page 7 to link to the next section.
- Page 8, line 239: please reconsider referring to people as 'remitted depressed' as personally it's not how I would like to be described and some readers may take offence (I am sure you don't intend any offence).
- Page 8, lines 262-264: given the very minimal detail you have provided about the included studies, it's hard to make sense of the information that is included. Here is an example where you talk about 21-day snapshot data from a 2-year study. Until now you have given no sense of how long any of the studies lasted for. Was this common in studies? Or was it just this study that used snapshot data? In this instances I am really wanting to see a table of included studies to make sense of the different types of designs, methodologies, and characteristics.
- Minor point (page 8, line 265): change 'was' to 'were'.
- Minor point (page 8, line 267): should 'worsens' be 'worsened'?
- Minor point (page 8, line 269): it's unclear to me why the phrase 'could be used' was chosen instead of 'were found to'. Was this your way of saying it has been proposed but not tested?
- Minor point (page 8, line 272): please specify what you mean by study spaces. To me, this means a private area where students engage in individual study (i.e., homework) but I may be incorrect (do you just mean educational setting?) and some clarity would help international readers.
- Minor point (page 9, line 273): regarding the frequency of launch per app, do you mean social media apps only? Or social media and social communication apps? Or any app?
- Minor point (page 9, line 278): reference [61] should be formatted as 61.
- Some reporting of the results seems inconsistent. Each time you include specifics (e.g., study characteristics or results) I feel as if you should do this systematically or not at all, as it's hard to understand the meaning. For example, on page 9 you offer percentages for classification accuracy and later on you offer descriptive categories (i.e., 'fair to good concordance') for screening accuracy. I'm not sure what constitutes fair or good accuracy. Later on (page 12, 398) you report a percentage of uptake by healthcare professionals, but I am unclear if this is across several studies or one study.
- Page 9, lines 297-299: perhaps this is in line with the nature of the review, but it felt out of place to

see unreferenced interpretation in the results section. I also felt that there were alternative explanations (such as in-person assessments allow for clarification about the nature of the questions, which can minimise overreporting).

- Page 10, line 324: you mention comparison groups, and I understand that based on the inclusion criteria that not all studies would have control or comparison groups, but without a table of included studies I am not sure how many did.
- Page 11, line 360: when you discussed usage statistics, I expected the references you cited to be systematic reviews or at least intervention studies. Have the correct references been inserted here?
- Page 11, line 364: did all YPN members unanimously decline to use RMTs? Or are they describing times when they have chosen not to (but they do at other times)? And why are there references for YPN comments? Are these concerns reflected in the studies you cite? If so, please make the alignment between YPN perspectives and the literature clear.
- Page 12, lines 391-394: here I am unclear whether these elements are considered important for cultural adaptation, or whether studies measured these as outcomes in some way to see if interventions could be successfully adapted for different cultural settings.
- Minor point (page 12, line 402): previously (and in the legend) you have used the acronym RMT to refer to technologies (i.e., plural), yet here you use it as singular. It would help readability if you used RMTs and RMT. Here I think you mean plural anyway, given that there are multiple different types (meaning on line 402 I would change 'its' to 'their').
- Minor point (page 12, line 412): change 'was' to 'were'.

Discussions and recommendations:

- The information about study duration, the tendency for studies to recruit interested participants, and use of community samples needs to be presented earlier in the review.

Recommendations for practice:

- I feel that the recommendation to avoid using RMTs as stand-alone interventions flows on well from the review findings. However, without having adequate information about the included studies, it is hard to see if there is sufficient evidence to support the use of blended, stepped-care approaches.

References: Does this manuscript reference previous literature appropriately? If not, what references should be included or excluded?

Yes.

Clarity and context: Is the abstract clear, accessible? Are abstract, introduction and conclusions appropriate?

Abstract:

- The abstract is mostly clear and accessible. Some sections could be edited for clarity, e.g., last sentence of background section of abstract.
- In the background section of the abstract you note that RMTs offer "objective" screening. This is a complicated claim given the diversity of things that RMTs measure (e.g., keyboard stroke), which require very subjective assessments to be made about what they mean. This is also true of the act of diagnosing depression, which is not an objective process.
- In the results section of the abstract, you note that smartphones were most 'preferred'; however, I wonder if it is more accurate to say that they were the most commonly studied. I also wonder if preference (or commonness) is best included in what 'works' or in another section.
- In the results section of the abstract, the sentence that reads "Adaptability was important, such that RMT were culturally compelling and accurate for the local context" doesn't tell the reader if this was important and done well, or if it was important but not done well.

- The conclusion section of the abstract lacks any detail. Rather than saying you have highlighted considerations, it would be much better to say what those considerations are. Rather than say you have made recommendations, it would be good to say what those are (and so forth). I understand the word limit makes this difficult, but given that most reviews will result in considerations, limitations/gaps, and recommendations, it would be better to have a few high-level sentences summarising the specifics here.

Please indicate any particular part of the manuscript, data, or analyses that you feel is outside the scope of your expertise, or that you were unable to assess fully.

None.

Kind regards,
Magenta Simmons.

Reviewer #3:

Remarks to the Author:

In this paper, the authors present a realist review of remote measurement technologies (RMTs) for depression in young adults aged 14 to 24 years. The topic is timely, and the importance of the topic is described in detail. I congratulate the team on their collaboration with the lived experience co-researchers from YPN. Overall, the paper was very interesting to read, and the realist approach was appropriate. I invite the authors to consider the following comments, which may help to further strengthen the review:

INTRODUCTION

1) Some more background information and discussion of the literature is required, to build a rationale for the realist review - has literature previously suggested that RMTs do not work for everyone, equally? Are there any existing programme theories on RMTs for depression in young people?

METHODS

2) Some readers may be unfamiliar with the realist methodology, so it would be helpful if the authors could add some more detail and explanation. For example, key to realist reviews is also the focus on mechanisms and unintended or negative outcomes. Additionally, realist reviews intend to produce a programme theory consisting of Context-Mechanism-Outcome configurations.

3) It is unclear whether any software was used to manage the literature. Please clarify how literature was managed. Were duplicates removed manually?

4) The methods are not entirely replicable. I would strongly encourage the authors to specify the keywords searched, in addition to providing more information on the data extracted (e.g., data extraction sheet), the constructs analysed and the specific analytic process.

RESULTS

5) Please provide some more information on the characteristics of the documents included in the review e.g., of the 104 items included, how many were grey literature, what types of studies were included, how many were protocols or reviews etc.

DISCUSSION

6) Acknowledgement of the strengths and limitations are missing. A common critique of realist reviews is that the programme theory is only as strong as the literature included in the review. Please consider acknowledging this (and the absence of formal quality appraisals) as a limitation.

Author Rebuttal to Initial comments

We thank the editors and the three reviewers for their time considering our manuscript and the important comments they have made, which we address as follows:

Reviewer 1

1. I found the description of the methods a bit opaque (e.g., lines 163 to 176). Most notably, it was not clear to me how the authors actually synthesized the literature. I think this is very important to clarify, as it seems a different research group could theoretically read the same literature and draw divergent conclusions. The review “felt” fairly balanced to me, but it would be very helpful to have details about how this synthesis occurred. I was particularly curious how the authors handled conflicting findings and null results. While these were discussed in some places, I am guessing there were many more instances that could not be fully represented. In addition, I think it would be worth including a table with information that was coded for each study.

The methods section now includes further detail on clarifying scope, search processes, literature selection criteria and appraisal, data extraction and evidence synthesis. We have also developed, “Supplementary Table 4 – Literature Selection, Appraisal & Data Extraction Form” (p.6) to exemplify these processes that were originally conducted in excel, with each table field a new column, and each record on a new row. Finally, we have also now included a table of all included records in the results section (Table 1, p.6), which describes the characteristics of each included record, as well as the main evidence extracted from each record, and more fully represents conflicting findings and null results.

2. Relatedly, it was also not clear to me how precisely the co-researchers’ perspectives impacted the study - what was changed / informed as a result of their inclusion? I appreciate the importance of including co-researchers with lived experience. However, lacking some specifics around this process and its impact, it can seem a bit like it was done for show rather than actually informing the final product. (I am not suggesting that is the case, but simply noting this potential impression.) I did appreciate the several places in the text where the co-researchers’ perspective was noted specifically. On the other hand, I do have some questions about drawing conclusions / overweighting the perspective of two individuals. It would be helpful to be more specific about which

opinions attributed to YPN were representative of the broader group vs. the co-researchers alone.

To help understand how the perspectives of the two young, lived experience co-researchers and YPAG members shaped the study, we have included the document that was circulated to them at the end of the study to complete the feedback circle as, “Supplementary Table 2 - Completing the Feedback Circle” (p.3).

3. Also related to methods, I was curious how the authors chose which evidence to privilege. From my perspective, it would make sense to privilege meta-analytic / systematic review evidence, following by randomized trials, etc. Did the authors apply some evaluation of the rigor of the reviewed studies in deciding which studies to discuss more?

Evaluation of the quality of evidence was conducted according to Pawson et al 2004 & 2005. Realist review methodology rejects the use of a privileging / hierarchical approach to study type, seeing merit in a wide range of evidence from diverse primary sources, which also makes it difficult to use standard appraisal checklists. Additionally, study type is an inappropriate aspect to focus on when in a realist review, only a single element of the study may be considered when identifying, testing and refining a specific theory. Instead, a realist review relies on judgement and discussion with stakeholders to establish whether the primary source is fit for purpose for identifying, testing and refining a specific theory. Specifically, we focused on primary sources that addressed particular theories under test (relevance) and had credible methodology and sufficient weight behind the inferences made (rigour), with assessment of the relative contribution of each primary source occurring during the evidence synthesis stage rather than as a separate process e.g., being cautious about what the extensive qualitative evidence for a positive opinion of RMT actually means in light of other evidence indicating young people wouldn't actually use RMT for their mental health in the real-world.

4. One broad question that arose for me is whether it is reasonable to think about RMT as an intervention. It seems that much of the available literature at this time is simply trying to determine whether depression can be assessed using RMT. Of course, eventually the goal is to have these tools translated into interventions, but is it reasonable to evaluate RMT as an intervention at this stage? From the perspective that RMT is not necessarily intended to be an intervention, some statements (e.g., RMT as “much-needed scalable solution to the global mental health crisis”, Line 383) seem a bit unreasonable.

This comment forms some of the reasoning behind why we decided to conduct a realist review on the use of RMT for depression in young people, as there are many publications that do suggest RMT as an intervention and the scalable solution to bridge gaps in mental health care (e.g., Rudd & Beidas 2020), covering everything from assessment, monitoring, symptom management, treatment and relapse prevention. Thereby creating a surge of interest and registered RCTs, when you are absolutely correct that the research and evidence base is still limited to whether depression can even be assessed using RMT, with limited research into

unintended outcomes and little regulation of their current use in mental health care – as highlighted in this realist review, and the recommendation not to currently view RMT as such.

Additionally, the realist review was conducted as part of the Wellcome Trust “Active Ingredients” 2021 commission, which describes active ingredients as interventions / aspects of interventions that make the difference in preventing, treating, and/or managing depression in young people i.e. intervention at any stage from early assessment to management to treatment.

5. The authors describe Fig 2 as a “realist intervention theory.” This may be primarily semantic, but I have some trouble seeing how this figure is a theory, per se. It seems to me more a visual representation / summary of the current state of the evidence with some theoretical pathways highlighted. I would suggest using different terms to describe this figure and not referencing a theory in the abstract.

Having reviewed the RAMESES Quality Standards for realist review (for researchers and peer reviewers) “4. Constructing and refining a realist intervention theory”, we feel that we have developed a realist intervention theory in practice but not presented it well in the figures in trying to save space and combining what was our initial framework with the realist intervention theory into one figure. As such, we have now included both our initial framework and the realist intervention theory as two separate figures, which should better show how we set out initial theories in a framework to populate with evidence, iteratively refined them as the review progressed and produced a realist intervention theory comprising context-mechanism-outcome (CMO) configurations and an explanation of the pattern of CMOs.

6. While much of the discussion was nuanced, some statements seemed overly broad. For example, Line 230 states that “RMT data features were able to distinguish those with depression from healthy controls.” However, the authors then go on to discuss several instances in which findings were null or inconsistent (e.g., various sleep parameters). I wonder if the general statements like that on Line 230 could be reconsidered for accuracy.

The broad, general statements have now been reworded to more accurately reflect the evidence discussed.

7. The authors conclude that the best use of RMT is within a blended, stepped-care approach (Line 468-470). However, it was not clear to me how precisely they arrived at this conclusion. It seems to me it may be premature to make recommendations for practice based on the state of the literature. Did they review studies that suggested RMT is useful within a blended, stepped-care approach? And, if not, what supports them making this conclusion?

We agree that since studies of the use of remote measurement technologies as part of a “blended, stepped-care approach” have not yet been conducted, we cannot conclude that this would be their current best use; however, it is important to show that this was the resounding consensus from young people with lived experience of depression for how they could be best used currently, and our conclusions have been adapted to reflect this (p.18).

8. Minor suggestions:

- Even though the authors did not conduct a meta-analysis, I think following the PRISMA guidelines would be helpful (e.g., including exact search terms, completing the PRISMA checklist, etc.).

Given that we followed RAMESES guidelines, now with the inclusion of a RAMESES checklist, we feel it would be disingenuous to also include a PRISMA checklist; however, we do agree it is necessary to include exact search terms, now found in “Supplementary Table 3 - Searches” (p.5).

- I was not sure what “worry in the absence of safeguarding” (Line 344) meant.

Wording has been updated to “health anxiety when viewing pertinent data without a healthcare professional present to clarify what it means” (p.11).

- Some parts of Figure 2 were not clear to me. For example, what did the arrow between active and passive mean? Were all the bullet points in the “Usage & Adherence” box related to lower usage and adherence? What did the bullet points in the “Accuracy” box mean - are these features that improve accuracy? A clear table note would help improve the interpretability, I think.

To make the figure more interpretable, it has now been separated into two figures – one showing the initial framework, and the other showing the realist intervention theory.

- On Line 460, the authors argue that depression outcomes need to be standardized. Don't we have widely available standardized measures of depression?

This referred to the use of different measures of adherence and standardized measures of depression across studies, making comparison across studies difficult, and the need for standardization in which measures we use e.g., Krause et al 2022 - International consensus on a standard set of outcome measures for child and youth anxiety, depression, obsessive-compulsive disorder, and post-traumatic stress disorder. Wording has been updated to better reflect this (p.17).

Reviewer 2

Key elements that were missing for me:

- I expected to see you state the theories more clearly and to show how they were refined by the end of the review and what influenced those refinements.

Having reviewed the RAMESES Quality Standards for realist review (for researchers and peer reviewers) “4. Constructing and refining a realist intervention theory”, we feel that we have developed a realist intervention theory in practice but not presented it well in the figures in trying to save space and combining what was our initial framework with the realist intervention theory into one figure. As such, we have now included both our initial framework and the realist intervention theory as two separate figures, which should better show how we set out initial theories in a framework to populate with evidence, iteratively refined them as the review

progressed and produced a realist intervention theory comprising context-mechanism-outcome (CMO) configurations and an explanation of the pattern of CMOs.

- A table of included studies that provided information about the study design, geographic location/s, setting, age/gender/ethnicity of participants, diagnostic information where available, etc.

We have now included a Table of all included records (Table 1, p.6). With a relatively large number of included records to describe, this is a complex Table and, as such, has been submitted as the separate excel file, "Table 1 - Included Literature"

(<https://www.nature.com/nathumbehav/submission-guidelines/aip-and-formatting#tables>).

Whole manuscript:

- Minor point: please check manuscript for long sentences that impede readability and could be broken into two sentences (there are some in each section, but the 'sleep' section on page 8 is a good example, as is the discussion of unintended consequences on pages 10-11).
- Optional suggestion: If you anticipate broad readership and you wish for the manuscript to be more understandable (e.g., by young people), you could consider including brief definitions for some of the technical jargon and phrases (e.g., sleep latency, time series data, accelerometry, classification accuracy, differences between emotional granularity, controllability, and self-regulation).

Long sentences have been broken down and brief definitions for technical jargon have been added.

Abstract:

- The abstract is mostly clear and accessible. Some sections could be edited for clarity, e.g., last sentence of background section of abstract.
- In the background section of the abstract you note that RMTs offer "objective" screening. This is a complicated claim given the diversity of things that RMTs measure (e.g., keyboard stroke), which require very subjective assessments to be made about what they mean. This is also true of the act of diagnosing depression, which is not an objective process (see related introduction comments & response below).
- In the results section of the abstract, you note that smartphones were most 'preferred'; however, I wonder if it is more accurate to say that they were the most commonly studied (both – smartphones were also most commonly studied, but this was likely related to the fact that acceptability studies indicated smartphones, particularly the individual's own smartphone, were most preferred). I also wonder if preference (or commonness) is best included in what 'works' or in another section.
- In the results section of the abstract, the sentence that reads "Adaptability was important, such that RMT were culturally compelling and accurate for the local context"

doesn't tell the reader if this was important and done well, or if it was important but not done well.

- The conclusion section of the abstract lacks any detail. Rather than saying you have highlighted considerations, it would be much better to say what those considerations are. Rather than say you have made recommendations, it would be good to say what those are (and so forth). I understand the word limit makes this difficult, but given that most reviews will result in considerations, limitations/gaps, and recommendations, it would be better to have a few high-level sentences summarising the specifics here.

Abstract significantly shortened to comply with the 150 word limit so some comments no longer apply.

Introduction:

- Some claims were missing a reference.

References added.

- The statement that diagnosing depression “can be challenging with a lack of objective, reliable diagnostic criteria...” feels out of place with the rest of the review (e.g., arguing for standardised measures of depression, referring to RMT measures as objective). The contradiction left me wondering what assumptions the authors have about the validity and reliability of depression diagnosis, and how these views potentially influenced the review.
- After you made the point about lack of reliable diagnostic criteria, I anticipated an interesting discussion about the tension of using RMTs to identify or predict something (i.e., diagnosis) that is associated with such challenges. When this didn't happen, I wondered if you instead meant that the criteria were reliable, but measures to assess whether someone meets these criteria are unreliable.

We intended, “lack of objective, reliable diagnostic criteria”, to mean that currently, there are no biomarkers for depression and that screening / diagnosis of depression relies on an individual's retrospective and subjective answers to structured clinical interviews and/or standardised measures of depression, which can be influenced by both recall and mood-state associated biases. With the ability to monitor biophysical changes and real-time measurements, use of RMT would certainly contribute to more objective screening / diagnosis of depression. However, we can see how our original sentence implies that there is currently no valid / reliable diagnostic criteria at all when there are many validated structured clinical interviews and/or measures of depression that certainly do work, so have rephrased this sentence (p.2).

- Minor point: given their recent emergence, it would be good to define ‘just-in-time adaptive interventions’ for readers in the first instance.

Removed just-in-time adaptive interventions from the introduction (p.2) and start of clinical utility section (p.10) as they fall under relapse-prevention.

- Page 3, line 108: you note that few clinical trials have been conducted, yet earlier you refer to this as a rapidly growing area of research. It may leave the reader wondering what type of research has been done if they are unfamiliar with the area. Including a sentence here would be useful. In the same sentence, it would be good to clarify if you are talking about effectiveness of RMTs in identifying or treating depression (or both).

We have now clarified what research has been conducted and what we mean by clinical effectiveness of RMTs (p.3).

Methods:

- Lived experience involvement is a strength of this review. It would be good to describe in general terms if lived experience involvement included people who had experienced use of RMTs for depression.

Inclusive, meaningful, and safe lived experience involvement is extremely important to us so we are greatly appreciative that you have highlighted this as a strength of the review. Given that the two young, lived experience co-researchers are included as named 2nd and 3rd authors, we were reluctant to mention specifics but note that there are more general terms we could use that still provide such clarity (i.e., “relevant lived experience”) and have updated the text (p.3)

- Inclusion criteria: can you please be more specific about the age range. Were studies included if the age of all participants was in the 12-24 age range? Or if the mean age was within the 12-24 age range?
- Exclusion criteria: please include brief justification of excluding specific types of depression.

The literature selection criteria has been updated with more detail to show how the inclusion and exclusion criteria were also refined during the iterative search processes.

- Data extraction and evidence synthesis: please see comment above (it would be really useful if you could include a figure or supplementary file of the initial intervention theories, how they changed over time, and what influenced these changes).

The methods section now includes further detail on clarifying scope, search processes, literature selection criteria and appraisal, data extraction and evidence synthesis. We have also developed, “Supplementary Table 4 – Literature Selection, Appraisal & Data Extraction Form” (p.6) to exemplify these processes that were originally conducted in excel, with each table field a new column, and each record on a new row. Finally, we have now included both our initial framework and the realist intervention theory as two separate figures, which should better show our initial theories, how they changed and the evidence that influenced these changes.

Results:

- Minor point: page 7, line 225, consider changing ‘is’ to ‘are’. **Noted & updated**
- Minor point: on page 7, line 230, do you mean to say ‘some RMT data features’? **Noted & updated**

- Minor point: some signposting is needed at the end of page 7 to link to the next section. **Further sub-titles added to the “What does and does not work?” section.**
- Page 8, line 239: please reconsider referring to people as ‘remitted depressed’ as personally it’s not how I would like to be described and some readers may take offence (I am sure you don’t intend any offence).

Apologies, I clearly had my “academic”, rather than “lived experience hat” on here – completely agree, and wording as been updated as such. (The number of records that mentioned “compliance” rather than “adherence” was not great to see either!)

- Page 8, lines 262-264: given the very minimal detail you have provided about the included studies, it’s hard to make sense of the information that is included. Here is an example where you talk about 21-day snapshot data from a 2-year study. Until now you have given no sense of how long any of the studies lasted for. Was this common in studies? Or was it just this study that used snapshot data? In this instances I am really wanting to see a table of included studies to make sense of the different types of designs, methodologies, and characteristics.

We have now included a Table of all included records (Table 1, p.6). With a relatively large number of included records to describe, this is a complex Table and, as such, has been submitted as the separate excel file, “Table 1 - Included Literature”

(<https://www.nature.com/nathumbehav/submission-guidelines/aip-and-formatting#tables>).

- Minor point (page 8, line 265): change ‘was’ to ‘were’. **Noted & updated**
- Minor point (page 8, line 267): should ‘worsens’ be ‘worsened’? **Noted & updated**
- Minor point (page 8, line 269): it’s unclear to me why the phrase ‘could be used’ was chosen instead of ‘were found to’. Was this your way of saying it has been proposed but not tested? **Updated to “have been suggested”.**
- Minor point (page 8, line 272): please specify what you mean by study spaces. To me, this means a private area where students engage in individual study (i.e., homework) but I may be incorrect (do you just mean educational setting?) and some clarity would help international readers. **Noted & updated**
- Minor point (page 9, line 273): regarding the frequency of launch per app, do you mean social media apps only? Or social media and social communication apps? Or any app? **Noted & updated**
- Minor point (page 9, line 278): reference [61] should be formatted as 61. **Noted & updated**
- Some reporting of the results seems inconsistent. Each time you include specifics (e.g., study characteristics or results) I feel as if you should do this systematically or not at all, as it’s hard to understand the meaning. For example, on page 9 you offer percentages for classification accuracy and later on you offer descriptive categories (i.e., ‘fair to good concordance’) for screening accuracy. I’m not sure what constitutes fair or good accuracy.

Unfortunately, this is a result of the current state of the literature, which inclusion of Table 1 should now help show, and why we call for standardisation of measures and/or reporting across studies.

- Later on (page 12, 398) you report a percentage of uptake by healthcare professionals, but I am unclear if this is across several studies or one study. **Noted & updated**
- Page 9, lines 297-299: perhaps this is in line with the nature of the review, but it felt out of place to see unreferenced interpretation in the results section. I also felt that there were alternative explanations (such as in-person assessments allow for clarification about the nature of the questions, which can minimise overreporting). **This was meant to be referenced and has been updated as such.**
- Page 10, line 324: you mention comparison groups, and I understand that based on the inclusion criteria that not all studies would have control or comparison groups, but without a table of included studies I am not sure how many did.

We have now included a Table of all included records (Table 1, p.6). With a relatively large number of included records to describe, this is a complex Table and, as such, has been submitted as the separate excel file, "Table 1 - Included Literature"

(<https://www.nature.com/nathumbehav/submission-guidelines/aip-and-formatting#tables>).

- Page 11, line 360: when you discussed usage statistics, I expected the references you cited to be systematic reviews or at least intervention studies. Have the correct references been inserted here?

In using the phrase "usage statistics", we were trying to differentiate between the concept of adherence measured in intervention studies / systematic reviews as referenced throughout the review, and likely "real-world" use of RMT outside of a study setting measured through the online surveys referenced.

- Page 11, line 364: did all YPN members unanimously decline to use RMTs? Or are they describing times when they have chosen not to (but they do at other times)? And why are there references for YPN comments? Are these concerns reflected in the studies you cite? If so, please make the alignment between YPN perspectives and the literature clear. **Noted and updated**
- Page 12, lines 391-394: here I am unclear whether these elements are considered important for cultural adaptation, or whether studies measured these as outcomes in some way to see if interventions could be successfully adapted for different cultural settings.

These are elements indicated by participants in qualitative aspects of acceptability and feasibility studies conducted in LMICs and wording has been updated to better reflect this.

- Minor point (page 12, line 402): previously (and in the legend) you have used the acronym RMT to refer to technologies (i.e., plural), yet here you use it as singular. It would help readability if you used RMTs and RMT. Here I think you mean plural anyway,

given that there are multiple different types (meaning on line 402 I would change 'its' to 'their'). **Noted and updated**

- Minor point (page 12, line 412): change 'was' to 'were'. **Noted & updated**

Discussions and recommendations:

- The information about study duration, the tendency for studies to recruit interested participants, and use of community samples needs to be presented earlier in the review.

Noted and now included under "Gaps in the Literature & Methodological Issues" as an additional subtitle in the results section (p.13 – 14).

Recommendations for practice:

- I feel that the recommendation to avoid using RMTs as stand-alone interventions flows on well from the review findings. However, without having adequate information about the included studies, it is hard to see if there is sufficient evidence to support the use of blended, stepped-care approaches.

We agree that since studies of the use of remote measurement technologies as part of a "blended, stepped-care approach" have not yet been conducted, we cannot conclude that this would be their current best use; however, it is important to show that this was the resounding consensus from young people with lived experience of depression for how they could be best used currently, and our conclusions have been adapted to reflect this (p.18).

Reviewer 3

Introduction:

- Some more background information and discussion of the literature is required, to build a rationale for the realist review - has literature previously suggested that RMTs do not work for everyone, equally? Are there any existing programme theories on RMTs for depression in young people?

Unfortunately, we were unable to add more background information and discussion of literature to the introduction for a number of reasons, including word count of the main text being limited to 5,000 words and movement of the first author to a non-academic institution limiting access to publications and the reference management software used. The manuscript also includes a specific section on the rationale for using realist review in the methods, as per the RAMESES checklist. Hopefully, inclusion of Fig.1 helps describe some of the existing theories on the use of RMT for depression in young people.

Methods:

- Some readers may be unfamiliar with the realist methodology, so it would be helpful if the authors could add some more detail and explanation. For example, key to realist reviews is also the focus on mechanisms and unintended or negative outcomes.

Additionally, realist reviews intend to produce a programme theory consisting of Context-Mechanism-Outcome configurations.

We have now included both our initial framework and the realist intervention theory as two separate figures, which should better show how we set out initial theories in a framework to populate with evidence, iteratively refined them as the review progressed and produced a realist intervention theory comprising context-mechanism-outcome (CMO) configurations and an explanation of the pattern of CMOs.

- It is unclear whether any software was used to manage the literature. Please clarify how literature was managed. Were duplicates removed manually?

Reference management software was used, and this has now been stated in the methods section (p.5).

- The methods are not entirely replicable. I would strongly encourage the authors to specify the keywords searched, in addition to providing more information on the data extracted (e.g., data extraction sheet), the constructs analysed and the specific analytic process.

The methods section now includes further detail on clarifying scope, search processes, literature selection criteria and appraisal, data extraction and evidence synthesis. We now also specify the keywords searched in “Supplementary Table 3 - Searches” (p.5). Finally, we have also developed, “Supplementary Table 4 – Literature Selection, Appraisal & Data Extraction Form” (p.6) to exemplify these processes that were originally conducted in excel, with each table field a new column, and each record on a new row.

Results:

- Please provide some more information on the characteristics of the documents included in the review e.g., of the 104 items included, how many were grey literature, what types of studies were included, how many were protocols or reviews etc.

We have now included a Table of all included records (Table 1, p.6). With a relatively large number of included records to describe, this is a complex Table and, as such, has been submitted as the separate excel file, “Table 1 - Included Literature” (<https://www.nature.com/nathumbehav/submission-guidelines/aip-and-formatting#tables>).

Discussion:

- Acknowledgement of the strengths and limitations are missing. A common critique of realist reviews is that the programme theory is only as strong as the literature included in the review. Please consider acknowledging this (and the absence of formal quality appraisals) as a limitation.

The discussion section now includes a discussion of the strengths and limitations of the realist review methodology.

Decision Letter, first revision:

23rd February 2023

Dear Dr. Walsh,

Thank you for your patience as we've prepared the guidelines for final submission of your Nature Human Behaviour manuscript, "A collaborative realist review of remote measurement technologies for depression in young people" (NATHUMBEHAV-22041033A). Please carefully follow the step-by-step instructions provided in the attached file, and add a response in each row of the table to indicate the changes that you have made. Please also address the additional marked-up edits we have proposed within the reporting summary. Ensuring that each point is addressed will help to ensure that your revised manuscript can be swiftly handed over to our production team.

We would hope to receive your revised paper, with all of the requested files and forms within two-three weeks. Please get in contact with us if you anticipate delays.

Nature Human Behaviour offers a Transparent Peer Review option for new original research manuscripts submitted after December 1st, 2019. As part of this initiative, we encourage our authors to support increased transparency into the peer review process by agreeing to have the reviewer comments, author rebuttal letters, and editorial decision letters published as a Supplementary item. When you submit your final files please clearly state in your cover letter whether or not you would like to participate in this initiative. Please note that failure to state your preference will result in delays in accepting your manuscript for publication.

In recognition of the time and expertise our reviewers provide to Nature Human Behaviour's editorial process, we would like to formally acknowledge their contribution to the external peer review of your manuscript entitled "A collaborative realist review of remote measurement technologies for depression in young people". For those reviewers who give their assent, we will be publishing their names alongside the published article.

Cover suggestions

As you prepare your final files we encourage you to consider whether you have any images or illustrations that may be appropriate for use on the cover of Nature Human Behaviour.

ORCID

Non-corresponding authors do not have to link their ORCIDs but are encouraged to do so. Please note that it will not be possible to add/modify ORCIDs at proof. Thus, please let your co-authors know that if they wish to have their ORCID added to the paper they must follow the procedure described in the following link prior to acceptance:

Nature Human Behaviour has now transitioned to a unified Rights Collection system which will allow our Author Services team to quickly and easily collect the rights and permissions required to publish your work. Approximately 10 days after your paper is formally accepted, you will receive an email in providing you with a link to complete the grant of rights. If your paper is eligible for Open Access, our Author Services team will also be in touch regarding any additional information that may be required to arrange payment for your article. Please note that you will not receive your proofs until the publishing agreement has been received through our system.

Please note that *Nature Human Behaviour* is a Transformative Journal (TJ). Authors may publish their research with us through the traditional subscription access route or make their paper immediately open access through payment of an article-processing charge (APC). Authors will not be required to make a

final decision about access to their article until it has been accepted. Find out more about Transformative Journals

[REDACTED]

Best regards,
Alex McKay
Editorial Assistant
Nature Human Behaviour

On behalf of

Charlotte Payne

Charlotte Payne, PhD
Senior Editor
Nature Human Behaviour

Reviewer #1:

Remarks to the Author:

I appreciate the opportunity to review this manuscript for a second time. I also appreciate the authors' efforts responding to the prior round of reviewer suggestions. I think the changes have improved the

manuscript. I continue to think this is an important topic and the size and scope of the literature reviewed here makes a valuable contribution. However, I still have some questions about the manuscript in its current form.

1. I think more needs to be said regarding the purpose of a realistic review and the value of this approach versus other evidence synthesis methods. The authors include a paragraph on this at the start of Methods, but I did not find this sufficiently clear or convincing. I realize they may not have space to go into details about this approach and its rationale in the main body of the manuscript, but a supplement discussing this would be very helpful. I think they should explicitly discuss precisely how various forms of evidence are synthesized and how “rigour” is considered. I think explicit discussion of the fact that some study designs (e.g., meta-analyses, randomized trials) are not privileged over others and the reason for this would be helpful.

2. I am having some trouble seeing how the realist review approach and the involvement of co-researchers and other stakeholders with lived experiences informed much of the Results section. As it stands, it seems to me it reads like a fairly standard narrative systematic review. There are a few places where the voices of the co-researchers / YPAG members are specifically called out, but many places where this did not occur. Perhaps I’m missing something in how this literature was reviewed?

3. I’m still struggling with the theoretical model figures. I still think a figure note could go a long way. For example, in the first theoretical figure, does the green box under “how and why” include mechanisms of benefit and the red box mechanisms of potential harm? I’m simply not sure how to interpret other parts of these figures. For example, what does “sensitivity and specificity” under “accuracy” mean to the reader? What does “characteristic underlying variability in usage, adherence, and accuracy” under “for whom?” mean to the reader? What do the question marks mean?

Reviewer #2:

Remarks to the Author:

Thank you for addressing the comments in full. I have no further suggestions other than one very minor point: the names of the supplementary tables do not correspond with the file names.

Reviewer #3:

Remarks to the Author:

The authors have made a great effort in revising the manuscript and the methods are now much more replicable. The review addresses an important topic and I believe that it is now suitable for publication.

Norina Gasteiger

Author Rebuttal, first revision:

We thank the editors and the three reviewers for their time considering our revised manuscript and responses, and address the outstanding points as follows:

Reviewer #1 (Remarks to the Author):

I appreciate the opportunity to review this manuscript for a second time. I also appreciate the authors' efforts responding to the prior round of reviewer suggestions. I think the changes have improved the manuscript. I continue to think this is an important topic and the size and scope of the literature reviewed here makes a valuable contribution. However, I still have some questions about the manuscript in its current form.

1. I think more needs to be said regarding the **purpose of a realist review and the value of this approach versus other evidence synthesis methods**. The authors include a paragraph on this at the start of Methods, but I did not find this sufficiently clear or convincing. I realize they may not have space to go into details about this approach and its rationale in the main body of the manuscript, but a supplement discussing this would be very helpful. I think they should **explicitly discuss precisely how various forms of evidence are synthesized and how "rigour" is considered. I think explicit discussion of the fact that some study designs (e.g., meta-analyses, randomized trials) are not privileged over others and the reason for this would be helpful.**

As per your and the editors suggestion, we have now included (as tracked changes):

- A paragraph in the introduction providing clearer justification of our choice of a realist review over other evidence synthesis methods.
- More detail in the methods and limitations sections explicitly discussing how various forms of evidence were synthesised and why.

2. I am having some trouble seeing **how the realist review approach and the involvement of co-researchers and other stakeholders with lived experiences informed much of the Results section**. As it stands, it seems to me it reads like a fairly standard narrative systematic review. There are a few places where the voices of the co-researchers / YPAG members are specifically called out, but many places where this did not occur. Perhaps I'm missing something in how this literature was reviewed?

The realist review approach and lived experience involvement informed the results in the ways listed below, which we've made clearer throughout the manuscript and as a specific sub-heading in the methods section (as tracked changes):

- Developing initial list of intervention theories for the use of RMT for depression in young people (exploratory searches, YPAG meeting) and shortlisting key intervention theories (2 co-researchers).
- Deciding search terms and inclusion criteria, helping to refine in light of emerging evidence (2 co-researchers).
- Selection, appraisal, extraction and evidence synthesis (2 co-researchers).
- Review of refined intervention theory and context-mechanism-outcome configurations – i.e., do they align with lived experience in the real-world (YPAG meeting and 2 co-researchers) and developing recommendations (2 co-researchers).

3. I'm still struggling with the theoretical model figures. I still think a **figure note** could go a long way. For example, in the first theoretical figure, does the green box under “how and why” include mechanisms of benefit and the red box mechanisms of potential harm? I'm simply not sure how to interpret other parts of these figures. For example, what does “sensitivity and specificity” under “accuracy” mean to the reader? What does “characteristic underlying variability in usage, adherence, and accuracy” under “for whom?” mean to the reader? What do the question marks mean?

Each figure now includes a more detailed legend, including definition of any acronyms used in the figure (as tracked changes). Additionally, the table from Figure 3 is now included as Table 2 in the manuscript.

Reviewer #2 (Remarks to the Author):

Thank you for addressing the comments in full. I have no further suggestions other than one very minor point: the names of the supplementary tables do not correspond with the file names.

Thanks for spotting and highlighting! The numbering of the supplementary tables should now correspond with those included in the manuscript and in file names.

Reviewer #3 (Remarks to the Author):

The authors have made a great effort in revising the manuscript and the methods are now much more replicable. The review addresses an important topic and I believe that it is now suitable for publication.

Thank you!

Final Decision Letter:

Dear Dr Walsh,

We are pleased to inform you that your Article "A collaborative realist review of remote measurement technologies for depression in young people", has now been accepted for publication in *Nature Human Behaviour*.

Please note that *Nature Human Behaviour* is a Transformative Journal (TJ). Authors may publish their research with us through the traditional subscription access route or make their paper immediately open access through payment of an article-processing charge (APC). Authors will not be required to make a final decision about access to their article until it has been accepted. Find out more about Transformative Journals

With best regards,

Charlotte Payne

Charlotte Payne, PhD

Senior Editor
Nature Human Behaviour